# *LARP4* mRNA codon-tRNA match contributes to LARP4 activity for ribosomal protein mRNA poly(A) tail length protection

Sandy Mattijssen[1], Aneeshkumar G Arimbasseri[2], James R Iben[1], Sergei Gaidamakov[1], Joowon Lee[1], Markus Hafner[3], Richard J Maraia[1,4]*

[1]Eunice Kennedy Shriver National Institute of Child Health and Human Development, National Institutes of Health, Bethesda, United States; [2]Molecular Genetics Laboratory, National Institute of Immunology, New Delhi, India; [3]National Institute of Arthritis and Musculoskeletal and Skin Diseases, National Institutes of Health, Bethesda, United States; [4]Commissioned Corps, US Public Health Service, Bethesda, United Staes

**Abstract** Messenger RNA function is controlled by the 3' poly(A) tail (PAT) and poly(A)-binding protein (PABP). La-related protein-4 (LARP4) binds poly(A) and PABP. *LARP4* mRNA contains a translation-dependent, coding region determinant (CRD) of instability that limits its expression. Although the CRD comprises <10% of LARP4 codons, the mRNA levels vary >20 fold with synonymous CRD substitutions that accommodate tRNA dynamics. Separately, overexpression of the most limiting tRNA increases LARP4 levels and reveals its functional activity, net lengthening of the PATs of heterologous mRNAs with concomitant stabilization, including ribosomal protein (RP) mRNAs. Genetic deletion of cellular LARP4 decreases PAT length and RPmRNA stability. This LARP4 activity requires its PABP-interaction domain and the RNA-binding module which we show is sensitive to poly(A) 3'-termini, consistent with protection from deadenylation. The results indicate that LARP4 is a posttranscriptional regulator of ribosomal protein production in mammalian cells and suggest that this activity can be controlled by tRNA levels.

DOI: https://doi.org/10.7554/eLife.28889.001

*For correspondence: maraiar@dir6.nichd.nih.gov

Competing interests: The authors declare that no competing interests exist.

## Introduction

A key control element of the stability and translatability of eukaryotic mRNA is the 3' poly(A) tail (PAT) which can vary from ~25 to 250 nucleotides (*Mangus et al., 2003*; *Eliseeva et al., 2013*), and accommodate multiple molecules of PABP (*Baer and Kornberg, 1980*). PAT length is associated with translation efficiency in early development (*Subtelny et al., 2014*), and for specific mRNAs in somatic cells (*Park et al., 2016*). PABP interacts with multiple different proteins involved in mRNA translation and stability (*Mangus et al., 2003*; *Ivanov et al., 2016*). Several of these proteins, among which are key factors involved in mRNA 3' exonucleolytic deadenylation, translation initiation and termination, share a similar peptide sequence termed PAM2 that interacts with the C-terminal domain of PABP (*Xie et al., 2014*). Some proteins with a PAM2 sequence including Paips 1 and 2, LARPs 1, 4 and 4B, also harbor other regions that interact with PABP (*Yang et al., 2011*; *Tcherkezian et al., 2014*; *Xie et al., 2014*; *Fonseca et al., 2015*). LARPs 1, 4 and 4B associate with translating polyribosomes (*Schäffler et al., 2010*; *Yang et al., 2011*; *Tcherkezian et al., 2014*; *Fonseca et al., 2015*).

**eLife digest** Genes are coded instructions to build proteins and other molecules that make up living organisms. To build a protein, the code within a gene is copied into a molecule called a messenger RNA (or mRNA short). The letters of the genetic code are then read in groups of three, referred to as codons, by a molecular machine called a ribosome. Each codon corresponds to one of the building blocks of all proteins, known as amino acids. However, because there are 64 possible codons but only 20 or so amino acids found in proteins, different codons can encode for the same amino acid.

In addition to determining the order of amino acids in a protein, the sequence of codons in an mRNA can have other effects too. Some sequences change how the mRNA binds with other molecules, while others affect how long the mRNA will last within the cell before it is broken down. In fact, the stability of an mRNA is an important way to control a gene's activity and genes that encode unstable mRNAs typically yield less protein.

Understanding the full information potential of the DNA code is a major goal of many biologists. Much research in this field has focused on single-celled organisms such as yeast, yet the regulation of mRNAs in yeast is generally less complex than it is in humans.

Mattijssen et al. have now asked how codons within the mRNA for a human protein called LARP4 affect the mRNA's stability. This protein binds to mRNA molecules, and the experiments uncovered a short segment of codons that made the mRNA of LARP4 very unstable. Replacing specific codons in this segment with other codons for the same amino acid caused the stability of the LARP4 mRNA to increase a lot. This in turn changed how much LARP4 protein was produced.

Amino acids are brought to their corresponding codons by molecules called transfer RNAs (or tRNAs for short). Mattijssen et al. found that the codons in the short segment of LARP4 mRNA that caused instability were matched by rare tRNAs. Increasing the levels of these low level tRNAs also increased how much LARP4 protein was produced. The elevated levels of LARP4 revealed a new activity for the protein. Almost all mRNAs have a so-called poly-A tail at one end, and the experiments showed that LARP4 binds to a range of mRNAs to help make these tails longer, which in turn makes the molecules more stable. Deleting the gene for LARP4 from mouse cells lead otherwise stable mRNAs to have shorter poly(A) tails and to become less stable. This includes the mRNAs that code for the proteins that make up ribosomes.

The regulation of the mRNAs that encode ribosomal proteins has been challenging to understand. These new results may reveal a network of signals that connects the amount of tRNAs in a cell to the production of ribosomes. Since ribosome production is central to controlling cell growth and division, these results may have broad implications in research into areas as varied as human development and cancer.

DOI: https://doi.org/10.7554/eLife.28889.002

The 'La module' of the eukaryote-ubiquitous nuclear La protein is comprised of a La motif (LaM) followed by an RNA recognition motif (RRM) that cooperate to form a RNA binding pocket that recognizes the extreme UUU-3'OH termini of RNA polymerase III transcripts and protects them from 3' exonucleases (*Bayfield et al., 2010*). The La module has been highly conserved by a few distinct La-related proteins (LARPs) that arose during eukaryotic evolution but diverged in other features of their structure and function (*Bousquet-Antonelli and Deragon, 2009*; *Maraia et al., 2017*). Of these, LARPs 1, 4, 4B and 6 are mostly cytoplasmic and associated with mRNAs (*Maraia et al., 2017*). Yet, except for vertebrate LARP6 which binds a highly conserved stem-bulge-loop found in the 5' UTRs of three α-collagen mRNAs (*Cai et al., 2010*; *Martino et al., 2015*; *Zhang and Stefanovic, 2016*), details of RNA binding by the La modules of LARPs 1, 4 and 4B, and how such binding may contribute to their activities are largely unknown (*Maraia et al., 2017*). The La module-containing N-terminal domain of LARP4 has been shown to bind homopoly(A) whereas its full length homolog, LARP4B has been shown to bind A-rich, U-containing RNA (*Yang et al., 2011*; *Küspert et al., 2015*), consistent with key differences in their La motifs (*Bayfield et al., 2010*; *Maraia et al., 2017*); their regulation also differs since LARP4 but not 4B mRNA is destabilized by TTP (*Mattijssen and Maraia, 2015*). LARP1 appears to bind poly(A) and to stabilize mRNAs containing the 5' terminal

oligo pyrimidine (5'TOP) motif (*Aoki et al., 2013*) which are comprised of mRNAs encoding ribosomal proteins (RP), translation factors, PABP and other proteins (*Meyuhas and Kahan, 2015*). LARP1 is known to have two RNA-binding domains, a La module in its N-terminal half (*Nykamp et al., 2008*) and a C-terminal HEAT motif that directly binds the 5' m7G cap and pyrimidine tract of the 5'TOP motif (*Lahr et al., 2017*). LARP1 may regulate a number of transcripts in addition to the 5'TOP mRNAs (*Blagden et al., 2009*; *Tcherkezian et al., 2014*; *Fonseca et al., 2015*; *Mura et al., 2015*).

Coding region determinants (CRD) of instability have been found in a small number of mRNAs, including *Fos* and *Myc* (*Lemm and Ross, 2002*; *Chang et al., 2004*) (reviewed in *Lee and Gorospe, 2011*) and for *Fos* involves interactions with PABP, translating ribosomes, and deadenylation (reviewed in *Chen and Shyu, 2011*). Accumulating evidence indicate that the overall fraction of optimal vs. suboptimal codons in a mRNA is a major determinant of mRNA decay in yeast (*Presnyak et al., 2015*). A potential link between mRNA codon use and decay appears more complex in higher eukaryotic cells in which the relatively high content of 3' UTR-destabilizing elements is a confounding issue (reviewed in *Chen and Shyu, 2017*). Unresolved issues include whether codon optimality plays a role in higher eukaryotes, and if so to what extent and to what degree do cellular tRNA dynamics including relative abundances as well as codon-anticodon restraints such as wobble decoding which slows translation elongation (*Stadler and Fire, 2011*) vs. direct Watson:Crick (W:C) decoding, contribute (*Chen and Shyu, 2017*).

We compared expression in HEK293 cells of the open reading frames (ORFs) encoding several proteins and found LARP4 to be uniquely low. Detailed characterization revealed a codon-specific, translation-dependent CRD of mRNA instability that comprises <10% of the LARP4 ORF that is a strong determinant of expression. We analyzed the unusual codon characteristics of this CRD and their matches to cellular tRNA levels which we determined for this study. Synonymous substitutions limited only to the CRD including wobble vs. W:C decoding were analyzed for effects on expression of full length LARP4 and correlations with cellular tRNA levels. The synonymous substitutions led to LARP4 expression levels over a >20 fold range with excellent correlation with tRNA levels and codon-anticodon restraints ($R^2$ = 0.9). Furthermore, mild to modest overexpression of the most limiting cellular tRNA cognate to CRD codons increased LARP4 levels in a dose-dependent manner. For some CRD constructs, this tRNA led to increased LARP4 production without increasing mRNA levels, while for other, more codon-optimized constructs, it increased LARP4 protein as well as the mRNA levels. Increases in LARP4 levels by either synonymous codon swaps or tRNA overexpression revealed its dose-dependent activity to promote longer PATs on heterologous mRNAs with associated stabilization. These results and poly(A) binding data that indicate 3' end-specific recognition and suggest protection from deadenylation, point to mechanisms by which LARP4 promotes mRNA stability and potential control of RPmRNA by tRNA levels.

## Results

### Identification of a coding region determinant in *LARP4* mRNA that limits expression

We cloned cDNAs for LARPs and La from their second codon to their stop codon into expression vector pFlag-CMV2 (*Figure 1a*) and transfected these into HEK293 cells with a plasmid encoding adenovirus VA1 RNA synthesized by RNA polymerase III as a control. LARP4 accumulated to much lower levels than any other, with LARP4B, a homolog of similar mass (below), as the highest (*Figure 1b*). Northern blotting showed that *LARP4* mRNA accumulated to the lowest levels (*Figure 1c*). VA1 RNA was increased by La due to direct binding, stabilization (*Rosa et al., 1981*; *Francoeur and Mathews, 1982*; *Mathews and Francoeur, 1984*) and longer half-life (not shown); although its levels were more similar with the other LARPs (*Figure 1c*). *LARP4* mRNA was less than LARPs 6 and 4B by ~100 and ~50 fold respectively (*Figure 1d*), likely reflective of different stabilities of their coding regions.

The F-LARP4 constructs (*Figure 1e*) were western blotted after transfection and exhibited the expected mobilities (*Figure 1f,g*). Fragments 1–286, 27–286 and 359–724 accumulated to higher levels than full length 1–724, and fragments 27–724 and 1–430 (*Figure 1f*). These data suggested a region within codons 287–358 as inhibitory to expression, which was confirmed by the internal

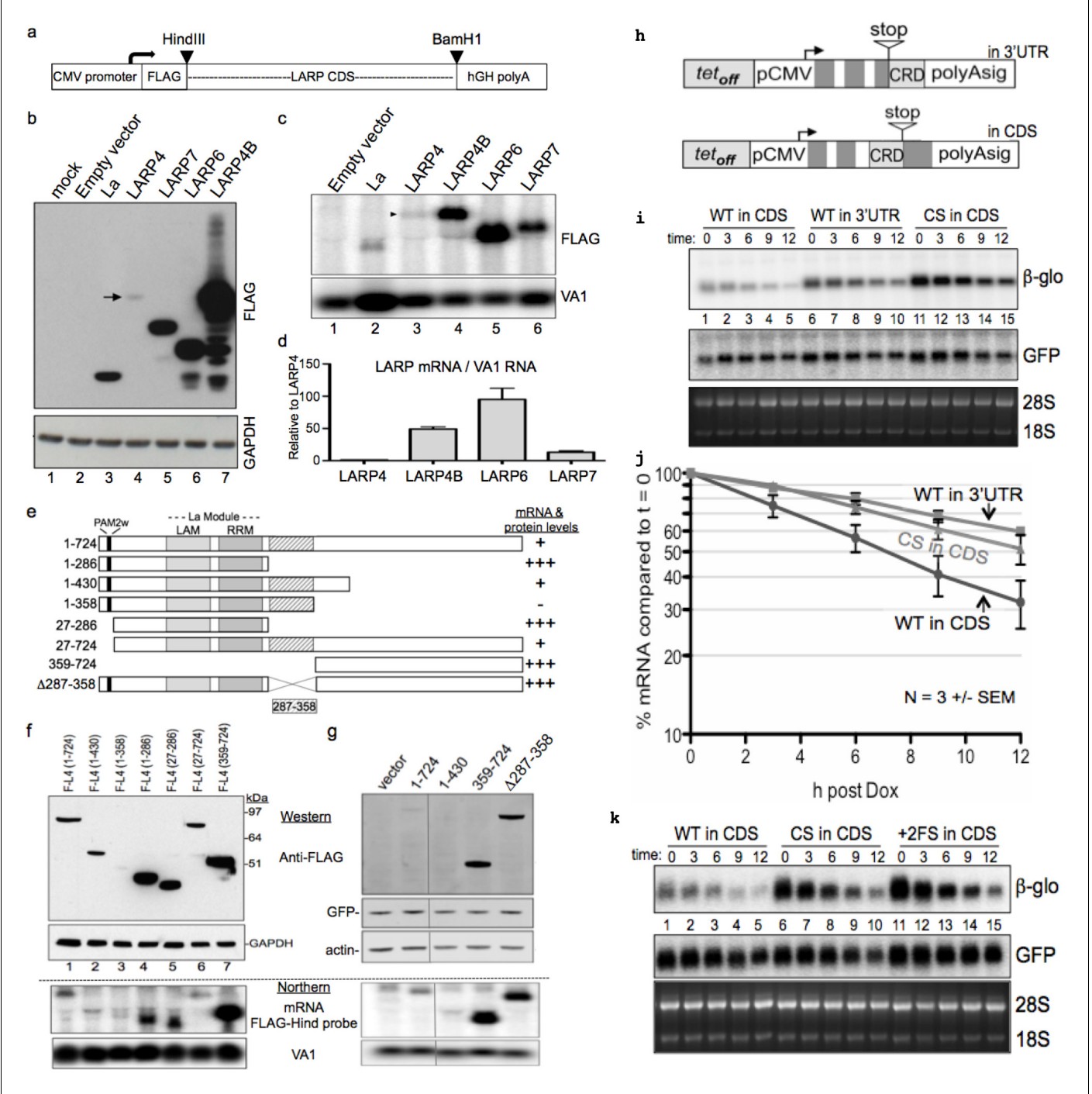

**Figure 1.** *LARP4* mRNA contains a codon-specific, coding region determinant (CRD) of instability in that limits expression. (a) Schematic showing expression cassette in pFLAG-CMV2 plasmids that differ only in the open reading frame (ORF) coding sequences (CDS). (b) Western blot using anti-FLAG antibody; arrow indicates LARP4 band, lane 4. (c) Northern blot, upper panel shows detection by FLAG-Hind antisense probe. VA1 RNA was for transfection control and quantification. (d) Normalized expression of three experiments (n = 3), error bars represent s.e.m.; Flag-LARP4 set to 1. (e) Schematic of LARP4 mutated constructs transfected with VA1 for f and g. (f) Upper: Western blot with anti-FLAG and anti-GAPDH. Lower: Northern blot with FLAG-Hind antisense and VA1 probes. (g) Upper: Western blot with anti -FLAG, -actin and -GFP antibodies. Lower: Northern for FLAG-Hind and VA1. (h) Schematic showing two β-globin reporters containing CRD constructs (see text); 'in 3'UTR' following the stop codon, and in frame preceding the stop 'in CDS'. (i) Northern blot time course of decay of the β-glo CRD reporter mRNAs. WT = wild type CRD sequence and CS = synonymous codon swapped version of the CRD. (j) Quantification of β-glo mRNA in i; 3 independent experiments, error bars represent s.e.m. GFP used for normalization, and each t = 0 was set to 100%. (k) Similar to i; +2 FS= + 2 frameshift version of the WT CRD.
DOI: https://doi.org/10.7554/eLife.28889.003

deletion construct, Δ287–358 that was expressed at levels nearly as high as 359–724 and much higher than 1–724 and 1–430 (*Figure 1g*). The differences in protein levels were generally reflected by the mRNAs (*Figure 1f,g* lower). Thus, codons 287–358 of *LARP4* mRNA contain a coding region determinant (CRD) that is inhibitory to expression. This comprises a tract of <10% of the ORF length that encodes part of LARP4 protein that is important for interaction with PABP, termed the PABP-binding motif (PBM) (*Yang et al., 2011*) (below).

## The LARP4 CRD mediates synonymous codon-sensitive mRNA decay

We used an established β-globin (βG) reporter under transcriptional control of a tetracycline/doxycy-cline-responsive promoter in HeLa tet-off cells to examine mRNA decay (*Gossen and Bujard, 1992*; *Fialcowitz et al., 2005*). The promoter is turned off upon addition of dox, and RNA is isolated at t = 0 and times thereafter to follow decay. During the 48 hr following transfection until t = 0, mRNAs transcribed at the same rate but with different half-lives accumulate to different levels, each requir-ing 3 to 4 half-lives to reach steady state (*Ross, 1995*). We inserted the LARP4 CRD into the βG-wt reporter in two contexts, in the ORF preceding the stop codon or following it (*Figure 1h*). When preceding the stop codon, the CRD led to t = 0 levels that were significantly lower as compared to placement after the stop (*Figure 1i* upper panel, lanes 1 and 6). Plotting triplicate time course data normalized to GFP mRNA showed that the CRD produced more instability as part of the ORF as compared to following it (*Figure 1j*, WT in CDS vs. WT in 3'UTR). The βG mRNA with the WT CRD in the CDS was indeed translated into a longer protein than with no insert (not shown). βG-wt with no insert yielded a half-life as expected (*Fialcowitz et al., 2005*) (not shown), similar to WT CRD in the 3'UTR.

We examined codon substitutions to the CRD. A synonymous codon-swapped (CS) CRD sequence inserted in the βG CDS increased reporter mRNA levels and half-life relative to the WT CRD in the CDS (*Figure 1i,j*). To further characterize the CRD and distinguish if destabilization might be due to RNA structure, G+C content, or codon-specificity, we inserted the WT CRD in the βG CDS beginning with a +2 frameshift (+2 FS) which required mutations to convert premature stops to sense codons. This preserved 97% CRD sequence identity but with only ~10% codon sequence iden-tity relative to the WT CRD. Similar to CS, the +2 FS largely reversed the inhibitory effect of the WT CRD (*Figure 1k*).

## LARP4 CRD is recognized as an inhibitory element when transferred to another mRNA

LARPs 4 and 4B share most amino acid and nucleotide homology in their La modules but less in other regions including the CRD which is only 52% nucleotide identical in this region (*Figure 2a*). We replaced the CRD region of LARP4B with the LARP4 WT CRD or a CS CRD and examined LARP4B expression. The WT-CRD decreased LARP4B-CRD-WT levels relative to LARP4B WT (*Figure 2b*, lanes 5, 4). Importantly, the CS CRD rescued the negative effect of WT-CRD (*Figure 2b*, lanes 6, 5); see quantitation in *Figure 2c*. Thus, the LARP4 CRD is recognized as inhibitory when transferred to a heterologous mRNA. When the test mRNA is normally expressed at much higher levels than LARP4, as in this case for LARP4B, the CRD appears to have less effect than on the lower abundance *LARP4* mRNA, but quantifications of duplicate experiments revealed that it nonetheless decreased expression of *LARP4B* mRNA by 2.5 to 3.5-fold (*Figure 2c*).

## The LARP4 CRD is active when followed by its natural regulatory 3'UTR of 4.2 kb

The native 3'UTR of *LARP4* mRNA is 4.2 kb and contains A+U rich elements controlled by TNFα via TTP (*Mattijssen and Maraia, 2015*). We replaced the short (0.48 kb) 3' UTR of pFlag-CMV2 LARP4 -WT and -CS with the LARP4 4.2 kb 3'UTR (*Figure 2d*). The long UTR lowered expression as expected (*Figure 2d*, lanes 2, 3; normalized to GFP). However, the CS CRD rescued the negative effect of the WT CRD (lanes 3 and 5, *Figure 2d*). These effects were reflected by differences in accu-mulation of the corresponding mRNAs containing the WT CRD and CS CRD in the long UTR (*Figure 2e*, upper panel, lanes 3 and 5). Quantification of data from duplicate experiments for the long and short mRNAs are shown in *Figure 2f*. Although the long UTR attenuated the effect as

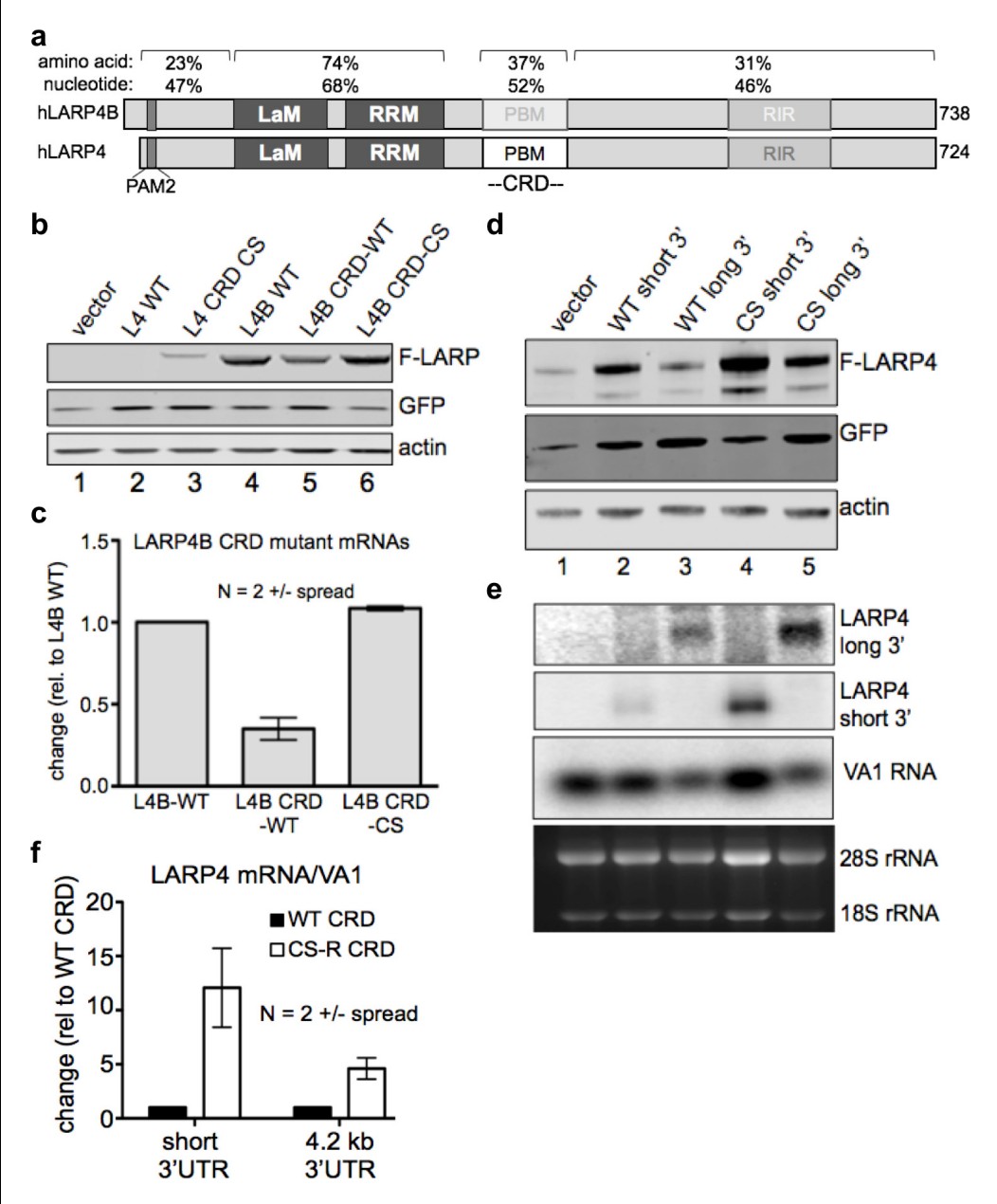

**Figure 2.** The LARP4 CRD confers instability when transferred to a mRNA of high expression level and in the context of its natural regulatory 3' UTR. (a) Diagram of human LARP4 and LARP4B with nucleotide and amino acid sequence identities of relevant subregions: PAM2 = PAM2, LaM = La Motif, RRM = RNA Recognition Motif, PBM = PABP interaction motif. (b–c) LARP4 CRD replaced the homologous region of LARP4B as either WT or the CS-B form (see text). Western blot b, and northern blot of F-*LARP4B* mRNA c, the latter quantified after two independent experiments; WT was set to 1 and normalization was to VA1, error bars = spread. (d-e) The 4.2 kb 3'UTR of LARP4 was inserted at the 3' end of the LARP4 WT or CS-R constructs (see below). Western blot d, and northern blot e. (f) Quantification of northern data from two independent experiments; error bars = the spread.
DOI: https://doi.org/10.7554/eLife.28889.004

compared to the short UTR, the WT CRD exerted a 5-fold decrease in mRNA levels in the context of its 4.2 kb 3'UTR relative to the CS CRD (*Figure 2f*).

## CRD synonymous codon substitutions that correlate with tRNA levels and decoding dynamics mediate >20 fold range in *LARP4* mRNA accumulation

In *E. coli*, yeast and some other organisms, tRNA gene copy number correlates with cellular tRNA abundance and codon use, whereas this correlation is low in humans (*dos Reis et al., 2004*). The tRNA adaptive index (tAI) is a measure of tRNA use by mRNAs that is derived from tRNA gene copy number and codon-anticodon base-pairing strength including that which distinguishes wobble vs. direct W:C pairing. Unlike in yeast, significant numbers of tRNA genes are variably inactive in different mammalian cell types and/or under different conditions (reviewed in *Orioli, 2017*). tAI scores are correlated with codon use and gene expression in yeast but not in human cells (*dos Reis et al., 2004*). Therefore, for the present study we determined tRNA levels in HEK293 cells by tRNA-Hydro-Seq (*Arimbasseri et al., 2015*; *Gogakos et al., 2017*) and the read counts for each tRNA species (*Table 1*) were used to derive cellular-tAI (ctAI) values. These values were incorporated into an algorithm that generated ctAI scores for mRNA ORFs relative to their optimal match to the HEK293 tRNA pool. Based on the tRNA read levels and this algorithm, we designed multiple additional CS constructs with synonymous mutations limited to the CRD region of full length LARP4, for comparison to WT and our original CS construct, hereafter designated CS-R. Designations and descriptions of the other CS constructs are as follows: CS-B was predicted to be expressed higher than CS-R, CS-W was predicted to be expressed lower than WT, and CS-I was predicted to be expressed at an intermediate level between WT and CS-R; the pattern of relative protein levels of these were generally as predicted (*Figure 3a*) and generally reflective of their mRNA levels (normalized to VA1, also see 28S RNA, *Figure 3b*). These results prompted more extensive detailed analyses (below).

Additional CS constructs in full length LARP4 were added to the above set and all were analyzed quantitatively for mRNA expression and correlation with the ctAI scores of their CRD regions (*Figure 3c*). The additional constructs differ from those in *Figure 3a* in a way that attempted to discern effects of wobble decoding which is prominent in the CRD (below). Constructs, CS-W, CS-I, CS-R and CS-B contain a mixture of synonymous codon swaps, some of which require decoding by a different tRNA anticodon than the original codon and some of which must be wobble decoded by the same tRNA anticodon as the original codon. Construct CSc contains synonymous swaps to Thr, Pro and Ile codons relative to WT, that require decoding by different tRNA anticodons. By contrast, constructs CSa, CSb and CS-Tyr contain synonymous swaps limited to U-to-C substitutions at wobble positions of select codons such that the same tRNA anticodon is used for decoding but with a direct anticodon G match to the wobble base C rather than wobble G:U decoding. CSb differs from WT by 14 synonymous codon swaps. Ten of these are weak codons, composed of all-A + U (UUU Phe and UAU Tyr) in the WT CRD. CSa differs from CSb only by five additional U-to-C substitutions at other weak codons in the CRD, at each of the Asn codons (AAU). Construct CS-Tyr differs from CS-B only at the 7 Tyr codons of the CRD; all of which are wobble UAU in CS-B, and in CS-Tyr they are UAC. For the Asn, Phe and Tyr codons, a single tRNA with G34 in the anticodon wobble position must decode both of their codons. Therefore, the U-to-C codon swap provides the only G:C base pair in these otherwise weak codons. The ctAI scores for the CRD regions of the constructs as well as the full length LARP4 constructs and some other proteins are in *Table 2*, and the sequences of CRD CS sequences are in *Figure 3—figure supplement 1*.

As can be seen in *Figure 3c*, quantitative analysis of mRNA expression levels by all of the LARP4 constructs revealed very good correlation with HEK293 cell tRNA levels and decoding dynamics as represented by the ctAI scores, with $R^2$ = 0.886. The range of expression obtained by the CRD synonymous swaps among all nine constructs tested, CS-W to CS-Tyr, spanned >20 fold (*Figure 3c*). Thus, although the CRD comprises <10% of LARP4 coding length, it is a significant determinant of its mRNA overall stability and translation.

## The LARP4 CRD exhibits complex codon clusters and bias for very low level tRNAs

The CRD is a *LARP4* mRNA feature that was localized by truncation and deletion constructs (see *Figure 1e*) and functionally analyzed thereafter. We developed a ctAI tool that calculates and plots a 10 nucleotide sliding window average translation proxy score along the length of an ORF. When applied to the human LARP4 sequence, the CRD appeared as a segment of high density low score

**Table 1.** tRNA read counts.

| tRNA # | Codon(s) | Anticodon | AA | AA | Read count | Total bin | | Fxn total |
|---|---|---|---|---|---|---|---|---|
| 1 | ACA/G/C | TGT | T | Thr | 6484 | | | |
| 2 | CCU/C | AGG | P | Pro | 6619 | | | |
| 3 | ACG | CGT | T | Thr | 7664 | | | |
| 4 | CCG | CGG | P | Pro | 9928 | | | |
| 5 | AUA | TAT | I | Ile | 11902 | | | |
| 6 | CAA | TTG | Q | Gln | 12533 | | | |
| 7 | CCA/G/U | TGG | P | Pro | 12777 | | | |
| 8 | AUU/C | AAT | I | Ile | 15657 | | | |
| 9 | ACU/C | AGT | T | Thr | 16261 | | | |
| 10 | AGC/U | GCT | S | Ser | 19618 | | | |
| 11 | UUU/C | GAA | F | Phe | 20453 | total bin1 | 139896 | 0.05387 |
| 12 | UUA | TAA | L | Leu | 23330 | | | |
| 13 | CUA | TAG | L | Leu | 23813 | | | |
| 14 | AGG | CCT | R | Arg | 24943 | | | |
| 15 | GGG | CCC | G | Gly | 25067 | | | |
| 16 | GUA | TAC | V | Val | 25169 | | | |
| 17 | UCG | CGA | S | Ser | 25390 | | | |
| 18 | CAU/C | GTG | H | His | 25607 | | | |
| 19 | GCG | CGC | A | Ala | 25778 | | | |
| 20 | CUU/C | AAG | L | Leu | 26991 | | | |
| 21 | UGG | CCA | W | Trp | 30456 | | | |
| 22 | UUG | CAA | L | Leu | 31930 | total bin2 | 288474 | 0.11109 |
| 23 | CAG | CTG | Q | Gln | 33162 | | | |
| 24 | UGU/C | GCA | C | Cys | 34487 | | | |
| 25 | CGG | CCG | R | Arg | 35475 | | | |
| 26 | GGA | TCC | G | Gly | 36754 | | | |
| 27 | CGA | TCG | R | Arg | 40154 | | | |
| 28 | UCA | TGA | S | Ser | 40530 | | | |
| 29 | UCU/C | AGA | S | Ser | 41241 | | | |
| 30 | AGA | TCT | R | Arg | 52949 | | | |
| 31 | GAA | TTC | E | Glu | 55795 | | | |
| 32 | GAU/C | GTC | D | Asp | 57771 | | | |
| 33 | CUG | CAG | L | Leu | 61112 | | | |
| 34 | CGU/C | ACG | R | Arg | 66635 | total bin3 | 556065 | 0.21414 |
| 35 | GCU/C | AGC | A | Ala | 74964 | | | |
| 36 | GGU/C | GCC | G | Gly | 77978 | | | |
| 37 | GUU/C | AAC | V | Val | 85733 | | | |
| 38 | GCA | TGC | A | Ala | 92573 | | | |
| 39 | AUG | CAT | M | Met | 102919 | | | |
| 40 | GAG | CTC | E | Glu | 113500 | | | |
| 41 | GUG | CAC | V | Val | 140741 | | | |
| 42 | AAA | TTT | K | Lys | 161366 | | | |
| 43 | UAU/C | GTA | Y | Tyr | 181682 | | | |
| 44 | AAU/C | GTT | N | Asn | 239876 | | | |

*Table 1 continued on next page*

*Table 1 continued*

| tRNA # | Codon(s) | Anticodon | AA | AA | Read count | | Total bin | Fxn total |
|--------|----------|-----------|-----|------|------------|-----------|-----------|-----------|
| 45 | AAG | CTT | K | Lys | 340378 | total bin4 | 1611710 | 0.62068 |
| | | | | Total | 2596700 | | | |

DOI: https://doi.org/10.7554/eLife.28889.008

clusters which otherwise occur relatively infrequently (*Figure 3d*) (apart from a stretch between residues 12–43 which may represent a conserved initial ramp of low codon optimality common to proteins within their first 50 amino acids [*Tuller et al., 2010*; *Shah et al., 2013*]). Fine mapping revealed that the lowest scoring points of LARP4 which ranged from 0.3 to 0.21 corresponded to two clusters of codons near the beginning of the CRD, denoted by red brackets at the left of the lower part of *Figure 3d*.

We sorted the HEK293 tRNA read counts into four bins (*Table 1*), each containing ~25% of the 45 tRNA anticodon species that decode the standard 61 sense codons (*Novoa et al., 2012*); bins 1, 2, and 4 contain 11 tRNAs and bin 3 contains 12. This revealed a wide range of tRNA levels; bins 1–4 comprise 5.4%, 11%, 21% and 62% of total read counts, respectively (*Table 1*). The numbers 1 to 11 above the codons in the lower part of *Figure 3d* represent the eleven least abundant tRNAs in the HEK293 cells, all in bin-1 (*Table 1*). The number 1 to 4 four lowest tRNAs, ThrUGU, ProAGG, ThrCGU, and ProCGG, ranged from 6500 to 10,000 reads (bin-1) and the four highest (bin-4) from 160,000 to 340,000 (*Table 1*). To verify a subset of these by another approach, semi-quantitative northern blotting confirmed that tRNAs ThrUGU, ProAGG and PheGAA as well as SerUGA (bin-3) were consistent with tRNA-Seq relative levels whereas TyrGUA appeared lower by northern which our data suggest may be due to base modification-mediated interference with probe hybridization (not shown).

Examination of the LARP4 CRD revealed multiple types of codon bias; only 33 of the 61 sense codons are found in the 71-codon long CRD (*Figure 3d*). 42% of all CRD codons must be decoded by bin-1 tRNAs which comprise only 5.4% of total tRNA abundance (*Table 1*). By contrast, 80% of codons excluded from the CRD are cognate to tRNAs in bins 2–4. Thus, the LARP4 CRD shows bias enrichment for codons cognate to low abundance tRNAs and bias for exclusion of codons cognate to high abundance tRNAs.

Strikingly, several bin-1 cognate codons are clustered. Moreover, many of these must rely on wobble decoding (red numbers, *Figure 3d*), which slows translation (*Stadler and Fire, 2011*). Some bin-1 clusters are flanked by or include other weak all-A +T codons (Tyr or Asn) that also require wobble decoding (*Figure 3d*, indicated by γ, λ).

## The LARP4 CRD is biased in weak wobble codons that are inhibitory to expression

As alluded to above, the genetic code contains seven sense codons with all-A-or-U nucleotides in the three positions (the other is a stop codon). HEK293 cells contain tRNA anticodon species to decode 4 of these 7 by direct W:C pairing whereas the other 3 must be wobble decoded. For each of these three, Asn AAU, Phe UUU, and Tyr UAU, the corresponding amino acids are encoded by only one other codon, that which ends with C in the wobble position and is W:C decoded by the single tRNA that must decode both codons. The CRD is biased in all 3 of the all-A+U wobble codons, Asn AAU, Phe UUU, and Tyr UAU, relative to their stronger synonymous codons, AAC, UUC and UAC. Specifically, 5 of 7 Tyr codons in the CRD are UAU, all 5 Phe codons are UUU, and all 5 Asn codons are AAU (*Figure 3d*), comprising NNU:NNC ratios of 5:2, 5:0, and 5:0 respectively. As the NNU:NNC ratios for these codon pairs range from 0.8:1 to 0.87:1 among all human ORFs (*Mauro and Chappell, 2014*), it is clear that the CRD is highly biased in its use of each of these three weak wobble codons.

The CS constructs that differ from their parent construct only in U-to-C wobble positions CSb and CS-Tyr, have significant effects on ctAI (*Table 2*). Therefore, the data show that U wobble codon decoding is suboptimal in the CRD because strengthening these codon:anticodon pairings appear to be functionally relevant. Comparison of the WT and CSb constructs (14 U-to-C wobble substitutions) revealed increase in LARP4 expression by 5-fold and this was further increased by additional

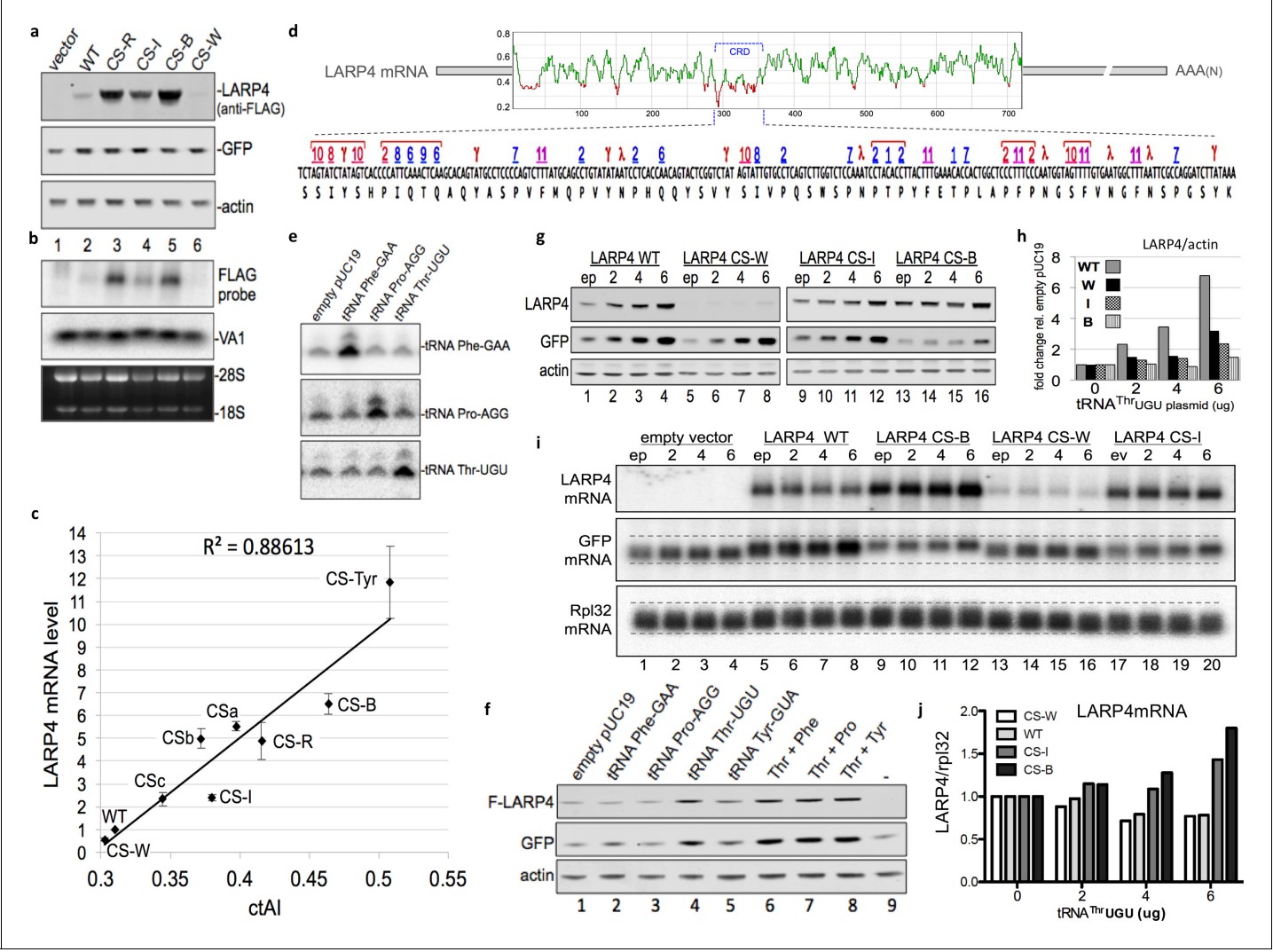

**Figure 3.** CRD synonymous codon match to limiting cellular tRNA levels and wobble dynamics control LARP4 expression levels. (**a**) Western blot of proteins from LARP4 codon swap (CS) constructs. WT = wild type LARP4; others are described in the text. (**b**) Northern blot of RNA from same cells in a. (**c**) Cellular tRNA index (ctAI) scores of the CRD regions of CS constructs. ctAI scores plotted on x-axis and Flag-*LARP4* mRNA levels relative to Flag-LARP4-WT on the y-axis; N = 7 biological replicates for CS-R and CSb, N = 5 biological replicates for CS-W, CS-I, and CS-B, N = 4 biological replicates for CSc, N = 3 biological replicates for CS-Tyr and CSa; error bars reflect the s.e.m. (**d**) Top: Sliding window ctAI-based score plot depiction of the full length LARP4 ORF of 724 codons (numbered on X-axis). Regions scoring below 0.4 (Y-axis) are colored red (see text). The position of the CRD codons 287–358 are indicated by the bracket. Bottom: The CRD codons are shown with single letter amino acids underneath and with numbers above that correspond to their cognate tRNAs as follows. Numbers 1–11 designate the rank order low level tRNAs in Bin-1 from *Table 1*; red = wobble decoded, blue = Watson:Crick decoded. Symbols γ and λ designate weak (all-A + T) codons that must be wobble decoded (see text) by non-Bin-1 tRNAs; note that #11 (F, Phe) is also a weak (all-A + T) codon. (**e**) Northern blot 48 hr after tRNA gene-containing plasmids were transfected into HEK293 cells. (**f**) Western blot of extracts from HEK293 cells transfected with empty pUC19 or plasmids containing tRNA genes for PheGAA, ProAGG, ThrUGU or combinations as indicated above the lanes, together with empty pCMV2 or F-LARP4-WT, and GFP. Antibodies used are indicated to the left of the panels. (**g–j**) HEK293 cells transfected with empty plasmid (ep) or 2, 4, or 6 ug of tRNA^ThrUGU plasmid together with empty pCMV2 (empty vector), F-LARP4-WT, CS-W, CS-I, or CS-B, and GFP. (**g**) Western blots as indicated. (**h**) Quantification of LARP4 in g as indicated. (**i**) Northern blot. (**j**) quantification of *LARP4* mRNA/Rpl32 mRNA from northern blot in i.

DOI: https://doi.org/10.7554/eLife.28889.005

The following figure supplements are available for figure 3:

**Figure supplement 1.** Sequences of the CRD regions of the LARP4 CS constructs.
DOI: https://doi.org/10.7554/eLife.28889.006

**Figure supplement 2.** Plots of 10 nt sliding window ctAI translation proxy scores of the full length LARP4 constructs.
DOI: https://doi.org/10.7554/eLife.28889.007

**Table 2.** ctAI scores for the CRD regions of LARP4 constructs (top) as well as the corresponding full length LARP4 constructs and some other reference proteins (bottom).

| CRD: | ctAI | Comments |
|---|---|---|
| WT | 0.3099 | |
| CSc | 0.3444 | Differs from WT by 13 synonymous substitutions that require decoding by different tRNA anticodons. |
| CS-I | 0.3802 | |
| CS-R | 0.4157 | |
| CS-B | 0.4638 | |
| CS-Tyr | 0.5082 | Differs from CS-B by U-to-C wobble substitutions at all 7 Tyr codons. |
| CSb | 0.3722 | Differs from WT by 14 U-to-C wobble substitutions (see text) but not Asn codons |
| CSa | 0.3973 | Differs from CSb only by U-to-C wobble substitutions at 5 Asn codons. |
| CS-W | 0.3028 | |
| **Full length:** | **ctAI** | |
| LARP4-WT | 0.4200 | |
| LARP4-CS-B | 0.4370 | |
| LARP4-CS-Tyr | 0.4409 | |
| LARP4-CS-W | 0.4190 | |
| LARP4B | 0.4286 | |
| LARP4B with LARP4 CS-B CRD | 0.4392 | |
| LARP4B with LARP4 CS-Tyr CRD | 0.4431 | |
| LARP4B with L4 WT CRD | 0.4227 | |
| hRpl35 | 0.6020 | |
| hActin (ACTG1) | 0.4883 | |
| hRps28 | 0.5190 | |
| eGFP | 0.5368 | |
| GAPDH | 0.4531 | |
| hRpl32 | 0.4866 | |
| H2A | 0.5347 | |

DOI: https://doi.org/10.7554/eLife.28889.009

Asn AAU codons to AAC in CSa. Separately, conversion of 7 Tyr UAU to UAC codons as reflected by CS-B vs. CS-Tyr, increased expression significantly (*Figure 3c*). We conclude that the LARP4 CRD is highly enriched in weak wobble codons and other codons cognate to very low level tRNAs, likely to slow ribosomes (*Stadler and Fire, 2011*; *Shah et al., 2013*) (Discussion).

Comparison of the ctAI plots of translation proxy scores of the synonymous codon swapped CRDs of the LARP4-CS constructs allowed visualization at near-codon resolution of the influence of wobble vs. direct W:C decoding (*Figure 3—figure supplement 2*). For example, higher scores mapped to Asn codons (asterisks, *Figure 3—figure supplement 2b*) in the CSa plot relative to CSb and also to the Tyr codons of the CS-Tyr plot relative to CS-B (*Figure 3—figure supplement 2b*).

## Overexpression of the most limiting CRD-cognate tRNA increases LARP4 expression

pUC plasmids containing human tRNA genes transfected into HEK293 cells led to 3–6 fold increases in the corresponding tRNAs (*Figure 3e*, and data not shown). When cotransfected with LARP4-WT and GFP, tRNA$^{Thr}$UGU increased LARP4 and GFP levels (*Figure 3f*) while tRNA$^{Phe}$GAA, tRNA$^{Pro}$AGG and tRNA$^{Tyr}$GUA did not (*Figure 3f*). Increase in GFP is consistent with a relatively high number of Thr codons in its mRNA and a limiting amount of cellular tRNA$^{Thr}$UGU. Because our data not shown indicated that different tRNA genes compete for expression, confounding the use of combinations thereof, we hereafter focused on overexpression of the single most limiting one, tRNA$^{Thr}$UGU.

*Figure 3g* shows western blots after transfection of HEK293 cells with increasing amounts of tRNA$^{Thr}$UGU plasmid (0, 2, 4 and 6 ug; empty plasmid, ep = 0 ug) along with LARP4-WT, CS-W, CS-I or CS-B. The basal levels of each LARP4 construct increased with increasing tRNA$^{Thr}$UGU and this also occurred for GFP (*Figure 3g*). An increase in GFP also occurred with LARP4-CS-W which is less active than LARP4-WT suggesting that the tRNA$^{Thr}$UGU effect is independent of LARP4 (corroborated in a later section). These data provide strong evidence that the low level of endogenous tRNA$^{Thr}$UGU is functionally limiting in these cells. Quantification of the response of the LARP4 constructs to tRNA$^{Thr}$UGU are shown in *Figure 3h* using the basal levels with empty plasmid set to 1. LARP4-WT exhibited the greatest response, up to a six-fold increase, while CS-W, CS-I and CS-B were less responsive (*Figure 3h*). This pattern suggests that WT LARP4 is programmed to be sensitive to limiting tRNA and that this reflects the unique composition of its CRD.

We also examined the effects of tRNA$^{Thr}$UGU on the LARP4 construct and GFP mRNAs as well as endogenous ribosome protein L32 (Rpl32) (*Figure 3i*). Remarkably, tRNA$^{Thr}$UGU did not increase LARP4-WT and CS-W mRNA levels (*Figure 3i*, lanes 5–8 and 13–16) despite the increase in their protein products (*Figure 3g,h*). This suggests that tRNA$^{Thr}$UGU increased the translational efficiency (protein/mRNA) of LARP4-WT, perhaps similar to that observed for HIS3 constructs with synonymous codons in yeast (*Presnyak et al., 2015*). By sharp contrast to LARP4-WT and -CS-W mRNAs, the tRNA$^{Thr}$UGU clearly increased the levels of LARP4-CS-B mRNA in a dose-dependent manner (*Figure 3i*, lanes 9–12), and to a lesser degree -CS-I (lanes 17–20). The quantifications are shown in *Figure 3j*.

The data in *Figure 3g–j* indicate that while overexpression of a single limiting tRNA can increase production of LARP4 protein from LARP4-WT mRNA, it does not lead to increased accumulation of this mRNA. This suggests that overcoming the destabilizing effects of the WT CRD with its intricate codon context (*Figure 3d*) may be too complex for resolution by a single limiting tRNA. Yet, LARP4-CS-B mRNA was increasingly stabilized by tRNA$^{Thr}$UGU and this was observed for CS-I although less so than for CS-B. We propose that because CS-I and CS-B CRDs contain more optimal synonymous codons than -WT and -W, they are more receptive to benefit from the accumulation effects of tRNA-$^{Thr}$UGU. These analyses indicate that the *LARP4* mRNA CRD can respond to a single limiting cognate tRNA with increased protein production, and moreover, that there may be a separate signal(s) in the CRD, apparently more complex than the Thr codons alone, that controls its instability determinant.

## Increasing tRNA leads to LARP4-dependent mobility shifts in heterologous mRNAs

Features in the patterns of GFP and Rpl32 mRNAs in *Figure 3i* are noteworthy because as will be shown in the next section they are relevant to LARP4 activity. *Figure 3i* revealed reproducible upward mobility shift of GFP mRNA, most readily appreciated by comparing lanes 4 and 5 and 12 and 13. The mobility shift was dependent on cotransfected LARP4 since lanes 1–4 did not show it. In addition, there was gradual but reproducible upward shift in the GFP mRNA band observable in lanes 5–8, 9–12 and 17–20 in response to increasing tRNA$^{Thr}$UGU. This was specific to cotransfected LARP4 because lanes 1–4 did not reveal it. We note that LARP4 CS-W did not reveal the upward shift of GFP mRNA probably because the levels of its LARP4 protein product were too low (*Figure 3g*). The GFP mobility shift is not due to electrophoresis or other artefact but as will be documented below results from a specific activity of LARP4.

The same general pattern observed for GFP was apparent for Rpl32 mRNA although less distinctly (*Figure 3i*). Another feature of Rpl32 mRNA is notable by comparing the vertical distribution of the bands within the lanes. Inspection of lanes 12 and 13 reveals the former more widely distributed than the latter, and similar for lane 5 vs. lane 4. As will be shown in the next and following sections, this reflects increasing PAT lengths.

## Elevated LARP4 levels increase its activity for mRNA PAT lengthening

The robust difference in the levels to which LARP4-WT and LARP4-CS-R accumulate revealed a dose-dependent shift in GFP mRNA relative to empty vector (*Figure 4a*). This also was observed for endogenous rpRpl32 (*Figure 4a*) as well as Rpl35, Rps27, and FAIM mRNAs (not shown, see below). The upward shifts also occurred with LARP4B (*Figure 4a*). GFP was more shifted than Rpl32 and other mRNAs (*Figure 4a*). Specifically, little of the shortest length GFP mRNA was present with

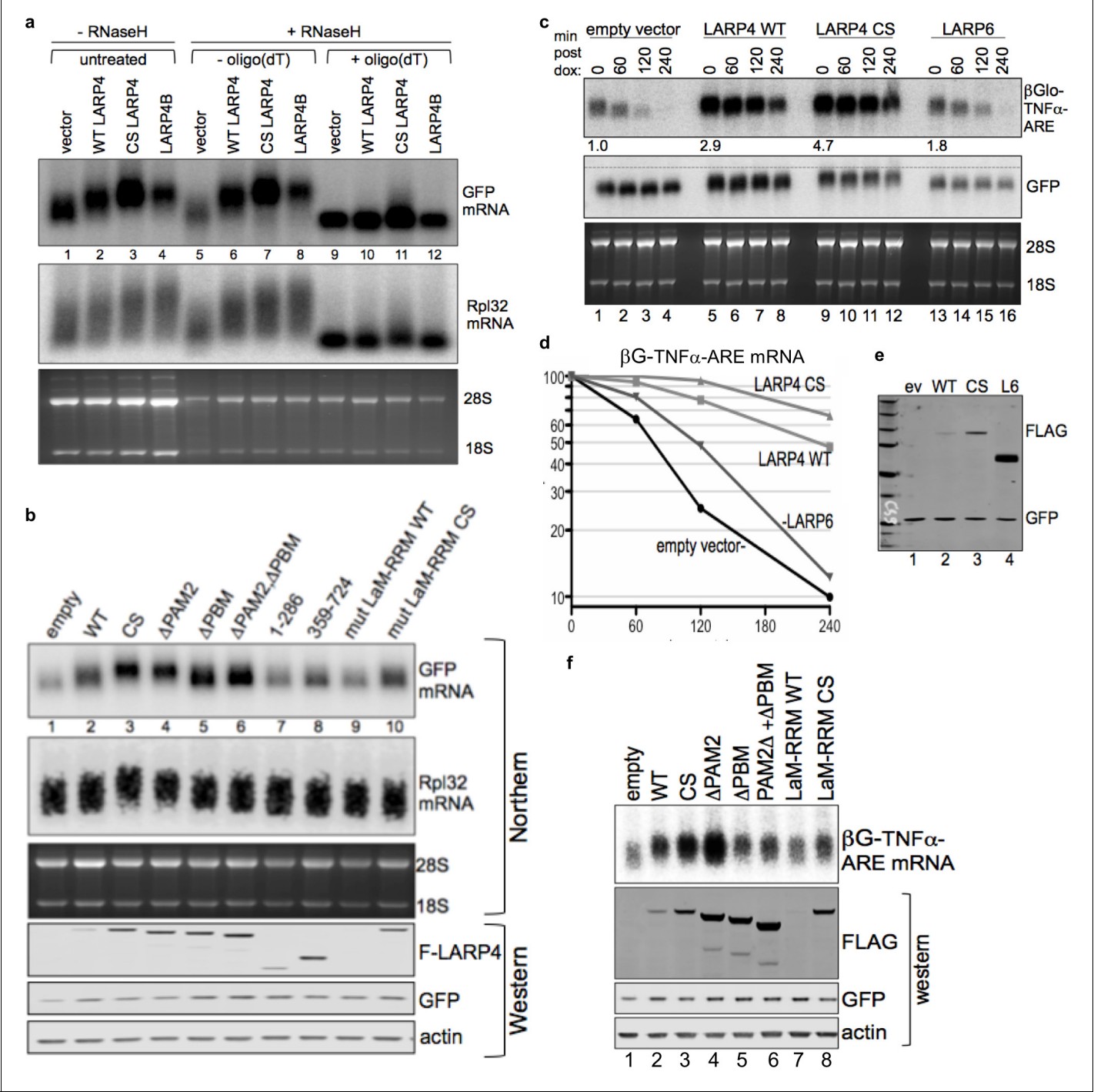

**Figure 4.** CRD-mediated increase in LARP4 leads to heterologous mRNA 3' PAT lengthening and stabilization dependent on its PABP- and RNA-interaction domains. (a) Northern blot after RNase H ± oligo(dT) treatment of total RNA from HEK293 cells transfected with constructs indicated above the lanes; CS = CS R version of the CRD in full length LARP4. (b) Upper: northern blot for GFP mRNA mobility shift activity of LARP4 constructs some of which contain the CS-R version of the CRD. WT = wild type, CS = full length LARP4 with CS-R CRD, ΔPAM2 = PAM2 deleted w/CS R, ΔPBM = PBM/CRD deleted, ΔPAM2ΔPBM, mut LaM-RRM WT and mut LaM-RRM CS = previously described M3 point mutations in the LaM and RRM in the full length LARP4 WT and CS-R versions respectively. Lower: western blot. (c) Northern blot of mRNA decay time course of HeLa Tet-Off cells transfected with βG-TNFα-ARE, GFP and either empty vector, LARP4-WT, LARP4-CS-R or LARP6. Cells harvested after 0, 60, 120 and 240 mins after doxycycline. Numbers under lanes for t = 0 indicate quantification of βG mRNA divided by GFP mRNA in the same lane, with lane 1 set to 1.0. (d) Quantification of βG-TNFα-ARE mRNA from c; the t = 0 for each was set to 100%. (e) Western blot of proteins tested in c, d. (f) Northern blot of HEK293 cells after transfection

*Figure 4 continued on next page*

*Figure 4 continued*

with βG-TNFα-ARE (constitutive promoter), GFP and either empty pCMV2 or F-LARP4-WT or mutants indicated above lanes as described for b; three lower panels show western blot of extracts using antibodies as indicated.

DOI: https://doi.org/10.7554/eLife.28889.010

LARP4 CS and LARP4B whereas short forms of Rpl32 mRNA remained. We believe this reflects that GFP mRNA PATs that were newly synthesized after transfection had not undergone shortening in the presence of ectopic LARP4, whereas shortened forms of cellular Rpl32 mRNA preexisted upon transfection with LARP4.

RNase H+ oligo(dT) treatment of RNA followed by northern blotting can reveal PAT length differences of specific mRNAs (*Shyu et al., 1991*) (*Figure 4a*, lanes 5–12). The mRNAs were converted to the same faster mobility form after cleavage by RNase H+ oligo(dT) (*Figure 4a*, lanes 9–12), indicating that their mobility differences were due to differences in the PATs.

We next examined the regions of LARP4 necessary for GFP mRNA PAT lengthening (*Figure 4b*, upper panel). LARP4 constructs that were mutated to debilitate binding to PABP by two motifs, PAM2 and PBM, as well as a mutant designated LARP4-M3 with point mutations to five residues in the LaM and two residues in the RRM of the La module had been described and characterized (*Yang et al., 2011*). Here we created the CS-R versions of those mutants that contained the CRD (*Figure 4b*, western). The upper panel of *Figure 4b* shows that LARP4 ΔPAM2 was partially active for PAT lengthening as evident by less GFP shift than LARP4 CS but more active than WT. ΔPBM exhibited less activity than ΔPAM2 whereas ΔPAM2-ΔPBM was similar to ΔPBM. Truncations 1–286 and 359–724, both lacking the PBM/CRD were qualitatively comparable to ΔPBM in shift mobility. The full length M3 LaM-RRM mutant disabled the shift activity in the low and high expression versions, WT and CS, lanes 9 and 10 respectively, consistent with its documented diminished association with PABP and polysomes (*Yang et al., 2011*). A similar trend was seen for Rpl32 and Rpl35 mRNAs (*Figure 4b*, and not shown) although as noted, their shifts are less distinct.

## LARP4-mediated PAT lengthening is associated with mRNA stabilization

We sometimes observed increased intensity of GFP mRNA signal with some LARP4 constructs without accompanying mobility shift, for example with the ΔPBM overexpressed proteins (*Figure 4b*). However, because a transcript requires 3–4 half-lives to achieve steady state (*Ross, 1995*), the relatively long lived GFP mRNA makes it unsuitable as an accurate reporter of stability for these transfection experiments. Therefore, we examined a β-globin (βG) mRNA reporter with a short half-life, βG-TNFα-ARE containing a destabilizing A+U rich (ARE) element from tumor necrosis factor (*Fialcowitz et al., 2005*).

HeLa tet-off cells were cotransfected with βG-TNFα-ARE, GFP, and the test plasmids that are indicated above *Figure 4c*. We included LARP6 as a control (*Ysla et al., 2008*). As indicated by the quantification values under the t = 0 lanes 1, 5, 9 and 13 of the top panel of *Figure 4c*, LARP4 CS led to more βG mRNA accumulation than LARP4 WT, LARP6 and empty vector. We also noted that the upper edge of the βG mRNA band shifted down after t = 0 with vector and LARP6, consistent with PAT shortening (*Ford et al., 1999*; *Lai et al., 2005*), but not with LARP4 WT and CS which maintained the longer forms (*Figure 4c*).

The data from the time course of βG mRNA decay were plotted in *Figure 4d* with the t = 0 values set to 100%. There was ~50% decrease in βG mRNA after 70 min with empty vector, in agreement with previous results (*Fialcowitz et al., 2005*). By contrast, mRNA stability was substantially increased by LARP4 as a decrease of 50% was observed at 240 min with LARP4 WT, and this was extended by LARP4 CS to ~65% remaining at 240 min (*Figure 4d*). The levels of the test proteins in this experiment are shown in *Figure 4e*.

As the GFP and Rpl32 mRNA mobility differences observed with LARP4-WT and -CS may reflect their relative activity levels for PAT lengthening (*Figure 4a*), a generally similar pattern was observed for their relative stabilization of βG-TNFα-ARE mRNA (*Figure 4d*). The t = 0 data indicated that LARP4-CS increased βG-TNFα-ARE mRNA levels ~1.6 fold relative to LARP4-WT; consistent with

this, extrapolation of the decay data suggested that LARP4-CS extended the mRNA half-life by ~1.5 fold relative to LARP4-WT (not shown) (*Ross, 1995*).

## LARP4 increases βG-TNFα-ARE mRNA accumulation in HEK293 cells

We wanted to analyze βG-TNFα-ARE expression in HEK293 cells in which tRNA dynamics were characterized but these cells do not have the tetracycline transactivator that could be used to shut off the promoter. We therefore cloned the βG-TNFα-ARE into a constitutive CMV expression plasmid and analyzed the reporter mRNA 48 hr after transfection, that is, comparable to t = 0 in the previous experiments. This allowed us to examine effects of LARP4 subregions on accumulation of βG-TNFα-ARE mRNA in these cells. This revealed that full length LARP4 WT and CS produced progressively more mRNA than empty vector consistent with the relative amounts of their protein products (*Figure 4f*). It is notable that the distribution of the βG mRNA in the empty vector was shifted upward by LARP4-WT at both its lower and upper edges (*Figure 4f*, compare lanes 1 and 2). The ΔPAM2 protein was expressed at higher levels than CS in this experiment and also led to higher βG mRNA levels (*Figure 4f*). By contrast, ΔPBM and ΔPAM2-ΔPBM produced significantly less βG mRNA than ΔPAM2 when expressed at comparable levels (*Figure 4f*); this is consistent with the PAT lengthening activity of the ΔPAM2 protein observed for GFP mRNA (*Figure 4b*). Thus, the PBM would appear to contribute more to PAT-mediated mRNA stability than does PAM2. Finally, the two M3 LaM-RRM mutants accumulated less βG-TNFα-ARE mRNA than their intact-LaM-RRM counterparts including when M3 LaM-RRM CS version was expressed as high as LARP4-CS (*Figure 4f*, lanes 3 and 8). We also note that the GFP mRNA appears to report more incremental changes in PAT length among different LARP4 constructs than does the βG-TNFα-ARE mRNA which may be due to the poly(A)-destabilizing effect of the ARE (*Ford et al., 1999*; *Lai et al., 1999*; *Lai et al., 2014*). We also observed LARP4-dependent upward shift of nanoluciferase mRNA, increased mRNA levels and increased nanoluciferase activity (not shown). The cumulative data suggest that LARP4 binds to poly (A) and PABP, and protects mRNA from deadenylation, resulting in apparent net increase in PAT length and stabilization.

## tRNA^ThrUGU overexpression increases LARP4 activity for mRNA PAT stabilization

We next examined HEK293 cells for effects of tRNA^ThrUGU on LARP4-mediated βG-TNFα-ARE mRNA accumulation. Cells were transfected with the βG-TNFα-ARE reporter, pUC19 (ep) or tRNA-^ThrUGU plasmid in combination with LARP4-WT, LARP4-M3 LaM-RRM mutant, or empty expression plasmid, and GFP, as indicated above the top panel of *Figure 5a*. In this experiment, tRNA^ThrUGU was increased 5–6 fold (*Figure 5a*). Protein levels are shown in *Figure 5b*. βG-TNFα-ARE mRNA was quantified in triplicate experiments (*Figure 5c*). Overexpression of tRNA^ThrUGU led to an upshift of βG-TNFα-ARE mRNA and increased its levels (*Figure 5a*, top) that was specific to LARP4-WT even though the LARP4-M3 LaM-RRM mutant level was increased by tRNA^ThrUGU as expected (*Figure 5b*). tRNA^ThrUGU stimulated a GFP upshift by LARP4-WT but not by LARP4-M3 LaM-RRM mutant (*Figure 5a*). The GFP shift was modest here because the increase in LARP4 levels by tRNA-^ThrUGU is not as much as the difference between LARP4-WT and LARP4-CS (*Figure 4a*).

tRNA^ThrUGU increased GFP protein independent of transfected LARP4 (*Figure 5b*, lanes 1, 2), corroborating that tRNA^ThrUGU is functionally limiting in these cells. Beyond this, the data validate the ability of LARP4 to promote mRNA PAT length and that this is associated with βG-TNFα-ARE mRNA stabilization. Further, the data show that increases in this novel LARP4 activity can occur in response to elevation of the level of a limiting cellular tRNA.

## Differential polysome profile distribution of *LARP4* mRNAs with synonymous CRDs in the presence and absence of tRNA^ThrUGU

We wanted to examine the effect of the CRD by comparing the distributions of LARP4-WT and LARP4-CS-Tyr mRNAs in HEK293 cell polysome gradient sedimentation profiles in the presence and absence of over-expressed tRNA^ThrUGU. To achieve similar levels of the LARP4 -WT and -CS-Tyr mRNAs we transfected less of the latter plasmid than the former, maintaining equal amounts of total transfected plasmid and GFP controls (methods). We first examined aliquots of the transfected cell extracts by western blotting (*Figure 6a*); F-LARP4 intensities in *Figure 6a* indicated that

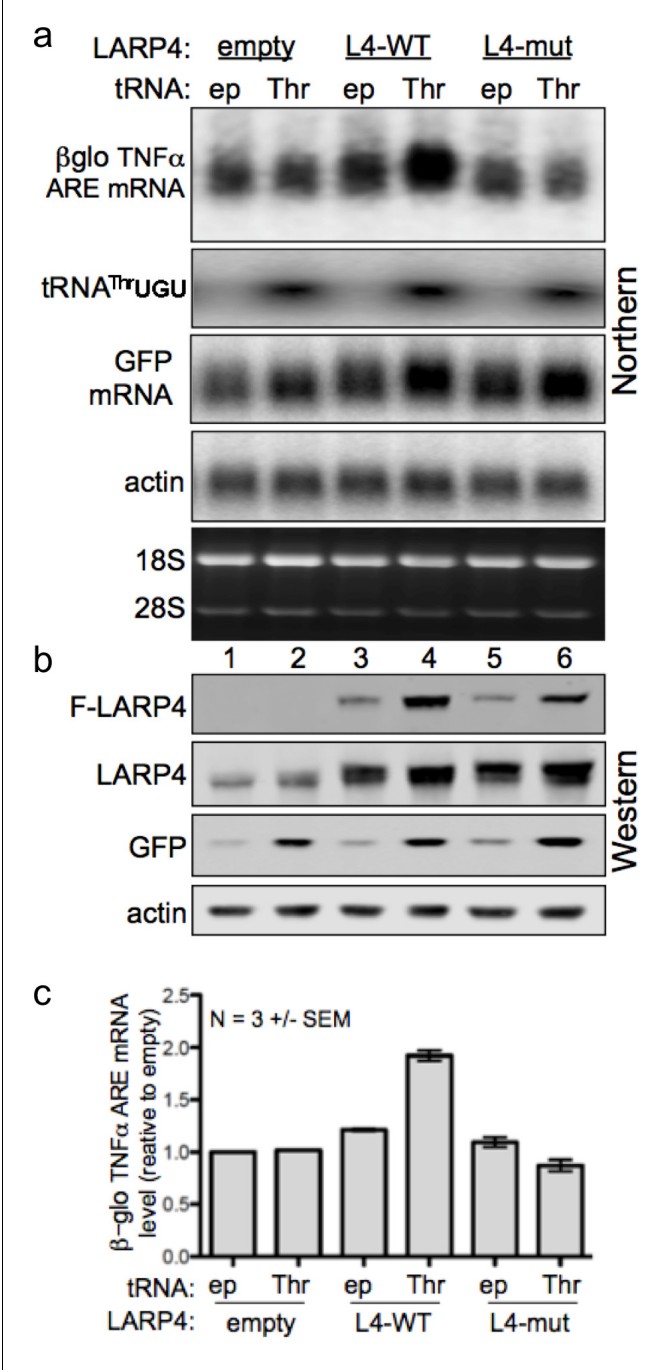

**Figure 5.** Increasing levels of the low abundance tRNA$^{Thr}$UGU elevates LARP4 with consequent PAT lengthening and stabilization of heterologous mRNAs. (**a**) Northern blot of HEK293 cells after transfection with βG-TNFα-ARE (constitutive promoter), GFP and either empty pUC19 (ep) or tRNA$^{Thr}$UGU plasmid together with empty pCMV2 or F-LARP4 (L4–WT) or L4-mut M3 containing mutated LaM-RRM. Other panels show probings for tRNA$^{Thr}$UGU, GFP mRNA and actin mRNA. (**b**) Western blot of extracts from the cells in a; anti-FLAG for top panel, anti-LARP4 for second panel. (**c**) Quantification of βG-TNFα-ARE mRNA in a from 3 independent experiments; error bars = s.e.m.
DOI: https://doi.org/10.7554/eLife.28889.011

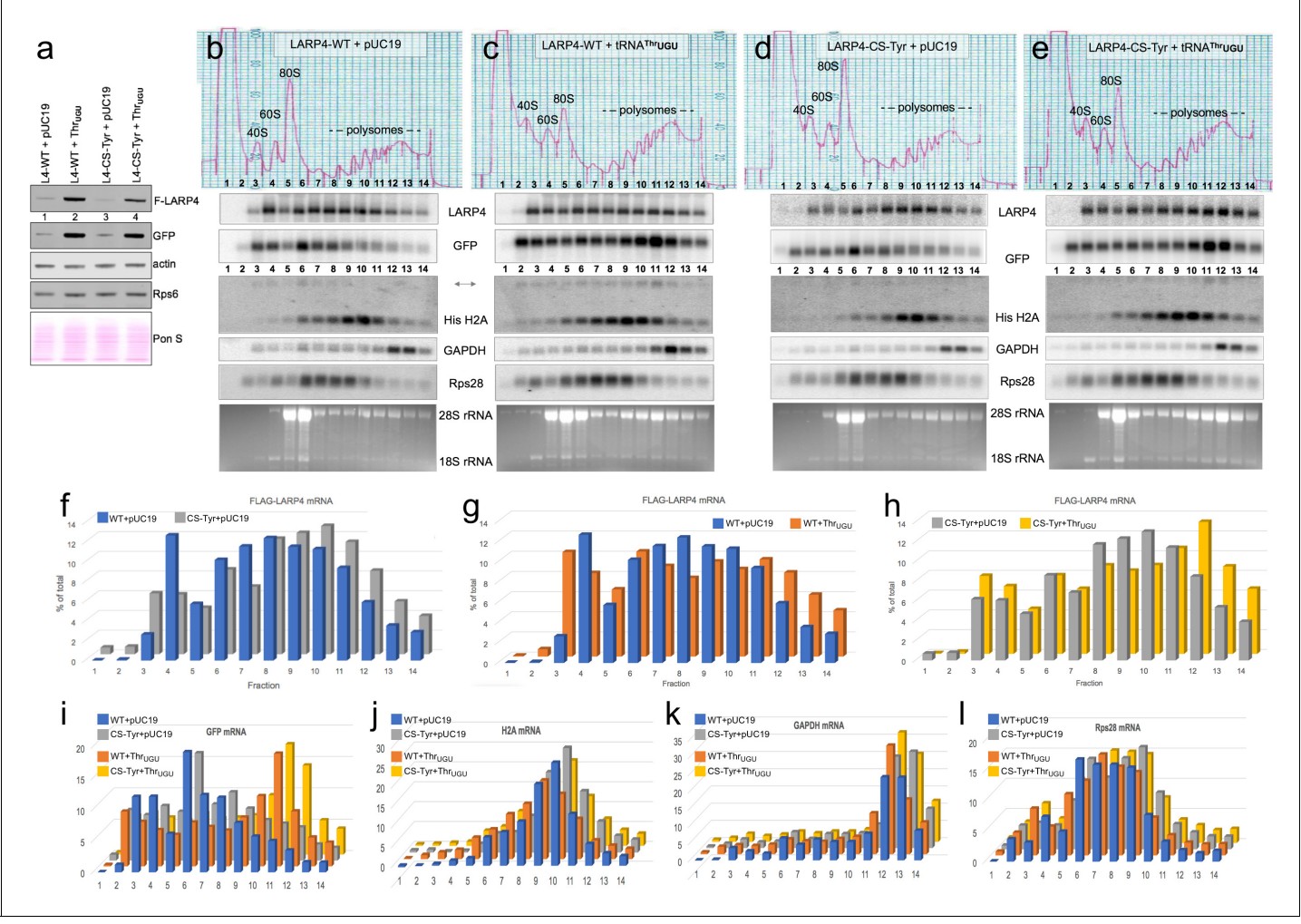

**Figure 6.** LARP4. Synonymous codon substitutions to the CRD and tRNA$^{Thr}$UGU promote apparent translation efficiency of LARP4 as monitored by polysome sedimentation analysis. (a) Western blot of HEK293 cells after transfection with GFP and either empty pUC19 or tRNA$^{Thr}$UGU plasmid together with F-LARP4 (L4–WT) or F-LARP4 CS-Tyr (L4-CS-Tyr) The total amount of plasmid transfected in each was the same in all four samples. (b–e) Polysome sedimentation profiles of the same extracts as in a. RNAs from each of the numbered fractions was analyzed by northern blot using the probes indicated next to the panels. The lower panels show ethidium bromide staining of the gels. (f–h) quantitations of FLAG-LARP4 from the northern blots in b-e. (i–l) Quantitations of different mRNAs in b-e as indicated above the graphs.

DOI: https://doi.org/10.7554/eLife.28889.012

The following figure supplement is available for figure 6:

**Figure supplement 1.** Time course in vitro translation of T7-synthesized mRNAs.
DOI: https://doi.org/10.7554/eLife.28889.013

tRNA$^{Thr}$UGU stimulated LARP4-WT production more than it stimulated LARP4-CS-Tyr, whereas it stimulated GFP more equally. In addition to the newly synthesized proteins, LARP4 and GFP, endogenous actin and Rps6 were examined, and the blotted membrane was also stained for total protein with Ponceau S to show relative loading (*Figure 6a*).

Polysome sedimentation profiles of the extracts were prepared in parallel and RNAs from each of the fractions were examined by northern blotting (*Figure 6b–e*). The distribution of a given mRNA species in a polysome sedimentation profile is determined in part by its rates of translation initiation, elongation and termination, as well as its overall length and codon length which limits the number of translating ribosomes. Several comparisons of the data collected are noteworthy. First, the polysome profile distributions revealed relatively more LARP4 -CS-Tyr than -WT mRNA in polysome fractions 8–14 than in fractions 3–7 (in the absence of tRNA$^{Thr}$UGU, *Figure 6b and d*), consistent with more efficient engagement of ribosomes by LARP4-CS-Tyr than LARP4-WT. The *LARP4* mRNAs in these

profiles were comparable in overall levels (*Figure 6b,d*); quantification is shown as the percentage of total mRNA in each fraction (*Figure 6f*). This revealed that a larger percentage of LARP4-CS-Tyr mRNA is in fractions 8–14 (66%) as compared to LARP4-WT (57%), providing evidence to suggest that CS-Tyr mRNA is occupied more densely by ribosomes than the -WT mRNA. This further suggests that synonymous codon substitutions to the CRD not only increase the levels of the LARP4-CS-Tyr mRNA (*Figure 3c*) but also its translational efficacy (Discussion).

The next comparison reflects tRNA$^{Thr}$UGU effects on the polysome distributions of LARP4-WT and LARP4-CS-Tyr mRNAs (*Figure 6b–e*). tRNA$^{Thr}$UGU shifted the LARP4 -WT and -CS-Tyr mRNAs toward heavier polysomes (*Figure 6g,h*). LARP4 contains 46 Thr codons in addition to those in the CRD; the LARP4 -WT and -CS-Tyr were shifted by tRNA$^{Thr}$UGU to different degrees (*Figure 6g,h*).

We note that tRNA$^{Thr}$UGU led to higher polysome levels and higher levels relative to 80S peaks in the OD254 tracings as compared to the control plasmid pUC19 for both LARP4 -WT and -CS-Tyr (top panels *Figure 6c* vs. b and e vs. d). This would not appear to be an artefact of polysome dissociation due to mishandling as reflected by comparable GAPDH mRNA profiles. For the mRNAs examined, the tRNA$^{Thr}$UGU and control blots were incubated with probes, washed and imaged together. Thus, the fractions from the tRNA$^{Thr}$UGU gradients (c and e) appear to contain more RNA than the pUC19 control gradients (b and d) as can be appreciated by comparing the H2A, Rps28, GAPDG and EtBr panels (*Figure 6b–e*).

A striking shift of GFP mRNA to heavier polysomes was observed in cells transfected with tRNA$^{Thr}$UGU plasmid (*Figure 6b–e,i*). It is important to note that *Figure 5* (and data not shown) indicate that tRNA$^{Thr}$UGU is more effective than LARP4-WT at increasing GFP protein levels (GFP western *Figure 5b* lanes 4 vs. 2) even though LARP4-WT increases GFP mRNA and its PAT length (*Figure 5a*), suggesting that tRNA$^{Thr}$UGU promotes its translational efficiency. This is consistent with the polysome distribution of GFP mRNA toward heavier polysomes in the presence of tRNA$^{Thr}$UGU (*Figure 6b & c and d & e*). Unlike LARP4, the GFP construct is far better codon optimized throughout its length for expression in human cells (*Haas et al., 1996*). Another consideration is that while the UTRs in the GFP and LARP4 constructs are comparably short, their ORF lengths differ at 240 and 724 codons respectively. Thus, there should be more potential homogeneity of ribosome-containing GFP mRNPs than LARP4 mRNPs. The more dramatic shift of GFP mRNA to heavy polysomes as compared to *LARP4* mRNA with tRNA$^{Thr}$UGU over expression may reflect both parameters, more benefit from codon context and more efficient ribosome occupancy per ORF length. In any case, the data provide evidence to indicate tRNA$^{Thr}$UGU as limiting for translation of newly transcribed mRNA from transfected plasmids in these cells.

We also probed for endogenous steady state mRNAs, graphically shown in *Figure 6j–l*. Both GAPDH and histone H2A were previously shown to be unaffected by ectopic LARP4 (*Yang et al., 2011*). The non-polyadenylated histone H2A ORF is 130 codons and the peak of its mRNA was mostly in the polysome fractions. GAPDH mRNA ORF is 336 codons and the peak of its mRNA was localized in the heavy polysome fractions reflective of efficient translation in all cases. For both H2A and GAPDH there appeared to be a slight shift to lighter polysomes in *Figure 6c* as compared to *Figure 6b* but less so for *Figure 6e* vs. *Figure 6d* (*Figure 6j,k*). The Rps28 ORF is 69 codons; its mRNA peak appeared to be more shifted to heavier polysomes by tRNA$^{Thr}$UGU for 6c vs. b and e vs. d (*Figure 6l*) as compared to H2A and GAPDH.

## LARP4 gene-deleted MEFs exhibit reduced RPmRNA PAT length and half-life

We produced LARP4 gene-deleted embryos from which knock-out (KO) and sibling wild type (WT) MEFs were made (*Figure 7a*). PAT length of RPmRNAs can be short (*Park et al., 2016*) which facilitates detection of net lengthening (*Figure 4a*), but may impede detection of further shortening. The mRNA lengths from KO and WT MEFs were analyzed for electrophoretic mobility by northern blots in triplicate (*Figure 7b*). Rps28 mRNA from KO migrated with focus toward faster mobility as compared to WT MEFs in which it was more widely distributed (*Figure 7b*, and right, lane tracings). Rpl32 mRNA showed a similar pattern although its ORF + UTR length is greater than Rps28 and not quite as well resolved (*Figure 7b*).

RpmRNAs are stable and accumulate to high levels relative to many other mRNAs. We also probed for protein phosphatase 1 regulatory inhibitor-14A (PPP1R14A, *Figure 7b*). The nonpolyadenylated histone H2A mRNA was comparable in KO and WT MEFs as expected (*Figure 7b*). Oligo

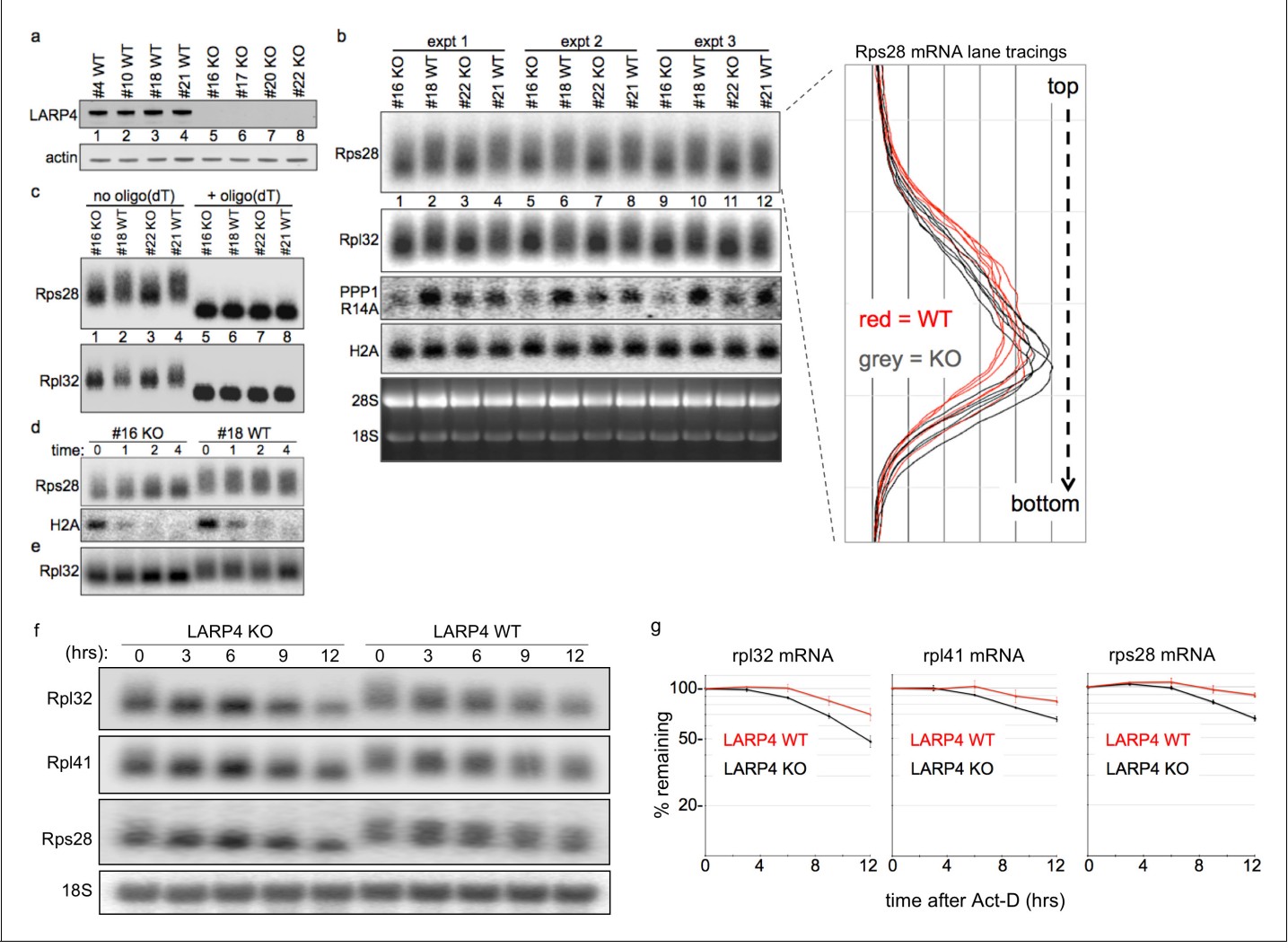

**Figure 7.** LARP4 gene-deleted knockout (KO) cells exhibit decreased 3' PAT length and stability of ribosome protein mRNAs. (a) Western blot of LARP4 from independent isolates of WT and LARP4 KO MEFs. (b) Northern blot from 4 MEF cell lines in a; probes as indicated to the left of the panels, Rps28, Rpl32, PPP1R14A, histone H2A mRNAs, and EtBr stained gel. Densitometric lane tracings for each lane of a Rps28 exposure is shown to the right as indicated. (c) RNase H assay in presence or absence of oligo(dT) as indicated. (d e,) Time course of mRNA decay in LARP4 KO and WT MEFs after transcription shut-off (in hrs) by actinomycin-D, probed for Rps28, Rpl32 and histone H2A mRNAs as indicated to the left; e contains the same RNA preparation as in d but run on a separate gel. (f) Northern blot of 12 hr act-D time courses for LARP4 KO and WT MEFs probed for the RNAs indicated to the left. (g) Graphs showing quantifications of duplicate experiments including panels in f, as indicated. The mRNA quantification at each time was normalized against 18S rRNA in the same lane. Error bars at each time point reflect the spread of the duplicates.

DOI: https://doi.org/10.7554/eLife.28889.014

(dT)-directed cleavage of poly(A) RNA produced similar fragments from KO and WT MEFs for Rps28 and Rpl32 mRNAs, indicating that the mobility differences in the absence of oligo(dT) are due to PAT length (*Figure 7c*).

We treated cells with actinomycin-D to block transcription, and isolated RNA at 0, 1, 2 and 4 hr thereafter (*Figure 7d,e*); short-lived H2A mRNA showed that act-D was effective. This revealed greater mobility differences for Rps28 and Rpl32 mRNAs in KO relative to WT MEFs, and more concentration in the shorter forms over the time course (*Figure 7d,e*). These data support a role for LARP4 in protection of mRNA PATs from 3' end shortening in vivo.

*Figure 7f* shows probings of a northern blot of a 12 hr time course after act-D treatment for Rpl32, Rpl41 and Rps28 mRNAs as well as 18S rRNA in LARP4 WT and KO MEFs. Later times were not examined because evidence of cell death was observed beyond 12 hr in act-D. Quantifications

are shown in *Figure 7g*, using the 18S rRNA for normalization. The data revealed that RPmRNAs decay faster in LARP4 KO than in WT MEFs (*Figure 7g*).

## Poly(A) 3'-end recognition by LARP4

A hallmark feature of the binding pocket of nuclear La protein (*Teplova et al., 2006*) is reflected by sensitivity of the RNA 3' terminal ribose 3'OH and 2'OH groups to chemical modifications (*Stefano, 1984*; *Terns et al., 1992*; *Dong et al., 2004*; *Teplova et al., 2006*; *Kotik-Kogan et al., 2008*). A previous study of LARP4 used a strong-hairpin RNA (*Yang et al., 2011*) which recent data suggest can interact with La module proteins via a binding mode that differs from single stranded RNA (*Martino et al., 2012*; *Martino et al., 2015*). We examined LARP4 for binding to single stranded A15 RNA with different 3' ends; 3'-OH, 3'-PO4, and 2'-O-CH3, and included U15 3'-OH as a sequence-specificity control. LARP4 showed highest avidity for A15 with 3'-OH, and progressively less for 3'-PO4, 2'-O-CH3 and U15 3'-OH (*Figure 8a*). Quantification is shown in *Figure 8b*. A second hallmark feature of the binding pocket of La for 3' end binding is sequence-specific recognition

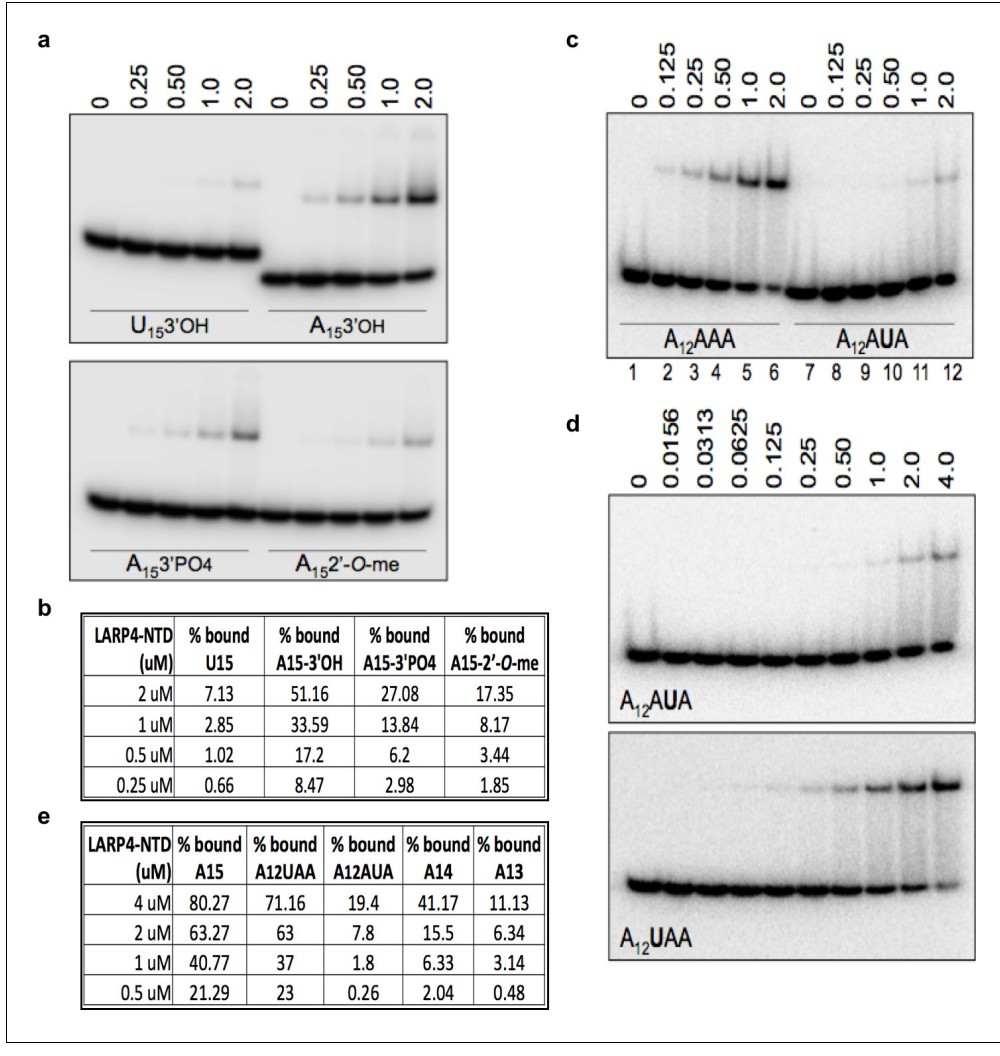

| LARP4-NTD (uM) | % bound U15 | % bound A15-3'OH | % bound A15-3'PO4 | % bound A15-2'-O-me |
|---|---|---|---|---|
| 2 uM | 7.13 | 51.16 | 27.08 | 17.35 |
| 1 uM | 2.85 | 33.59 | 13.84 | 8.17 |
| 0.5 uM | 1.02 | 17.2 | 6.2 | 3.44 |
| 0.25 uM | 0.66 | 8.47 | 2.98 | 1.85 |

| LARP4-NTD (uM) | % bound A15 | % bound A12UAA | % bound A12AUA | % bound A14 | % bound A13 |
|---|---|---|---|---|---|
| 4 uM | 80.27 | 71.16 | 19.4 | 41.17 | 11.13 |
| 2 uM | 63.27 | 63 | 7.8 | 15.5 | 6.34 |
| 1 uM | 40.77 | 37 | 1.8 | 6.33 | 3.14 |
| 0.5 uM | 21.29 | 23 | 0.26 | 2.04 | 0.48 |

**Figure 8.** The N-terminal LaM-RRM module-containing fragment of LARP4 exhibits poly(A) 3' end sensitivity. (**a**, **c** and **d**) Electrophoretic mobility shift assay (EMSA) using purified recombinant LARP4-NTD (1-286) protein in varying concentrations indicated above the lanes in uM with the purified oligo-RNA species indicated in the lower part of the gels. (**b** and **e**) Quantification of binding in a, c and d, and EMSA experiments with other oligo-RNAs as indicated.

DOI: https://doi.org/10.7554/eLife.28889.015

of the penultimate nucleotide which when substituted leads to loss of overall affinity (*Teplova et al., 2006*; *Kotik-Kogan et al., 2008*). *Figure 8c,d* indicate that LARP4 is sensitive to the penultimate A, i.e., at position minus-2 (A12AUA), significantly more so than at −3 (A12UAA); quantification in *Figure 8e*. While other features of RNA binding by La and LARP4 clearly differ in sequence specificity and RNA length requirement (*Yang et al., 2011*) and this remains to be understood at the structural level (reviewed in *Maraia et al., 2017*), the data in *Figure 8* demonstrate sensitivity of LARP4 to the poly(A) RNA 3′ end in a manner similar to the La module RNA binding pocket of La protein during RNA 3′ end sequestration (*Teplova et al., 2006*) and are consistent with a proposed mechanism for LARP4 for mRNA 3′ PAT protection from deadenylation.

## LARP1 also exhibits mRNA PAT length activity

LARP1 is known to bind, stabilize and regulate the translation of RPmRNAs (*Tcherkezian et al., 2014*; *Fonseca et al., 2015*; *Lahr et al., 2017*). It was reported that LARP1 could recognize the extreme 3′ terminus of poly(A) with sequence specificity for 3′ A, in an extract based system (*Aoki et al., 2013*), and directly bind PABP (*Blagden et al., 2009*; *Burrows et al., 2010*; *Tcherkezian et al., 2014*; *Fonseca et al., 2015*). We found that LARP1 produced a βG-TNFα-ARE length shift (*Figure 9a*) and a GFP mRNA shift comparable to LARP4 -WT and -CS (*Figure 9b*, lanes 1–4). RNase H + oligo(dT) demonstrated that this was due to net increase in mRNA 3′ PAT length (*Figure 9b*). Histone H2A mRNA exhibited insensitivity to RNase H + oligo(dT) as expected (*Figure 9b*). The proteins expressed in this experiment are shown in *Figure 9c*.

## Discussion

The data reported here show that human *LARP4* mRNA contains a translation-dependent coding region determinant (CRD) of instability that limits accumulation of the mRNA and the protein in HEK293 cells. This CRD is centrally located and represents less than 10% of the *LARP4* mRNA ORF. By overcoming the negative influence of the CRD, either by elevating the levels of a very low abundance limiting tRNAThrUGU in HEK293 cells or by synonymous codon substitutions to the CRD, LARP4 levels were increased. This revealed its new cellular activity, to mediate PAT length and associated stabilization of heterologous mRNA. This PAT net lengthening activity was shown for naturally stable RPmRNAs as well as a reporter mRNA with an ARE that is known to mediate deadenylation (*Ford et al., 1999*; *Lai et al., 1999*; *Lai et al., 2014*). This activity requires the intact PABP-interaction motifs of LARP4 and its intact RNA binding La module. Moreover, the results also suggest that this LARP4 activity is responsive to cellular tRNA levels.

## The LARP4 CRD is a modular, codon-specific, mRNA instability element

It is remarkable that *LARP4* mRNA levels were ~50 fold lower than LARP4B (*Figure 1d*) and that a substantial part of this could be rescued by synonymous codon substitutions to the CRD which comprises less than 10% of the coding region (*Figure 3c*). Other data demonstrated that the LARP4 CRD conferred significant instability when transferred to the higher level *LARP4B* mRNA (*Figure 2b, c*). It was recently noted that while much understanding of codon optimality has come from studies of yeast, outstanding issues include whether codon optimality plays a major role in mRNA decay in higher eukaryotic cells whose 3′ UTRs tend to contain a plethora of regulatory elements (*Chen and Shyu, 2017*). Our data showed that the LARP4 CRD was a significant instability element in the context of the *LARP4* mRNA 4.2 kb 3′ UTR (*Figure 2d–f*), which was previously documented to harbor negative regulatory elements responsive to TNFα and TTP (*Mattijssen and Maraia, 2015*).

Results from yeast indicate that codon optimality is a major determinant of mRNA stability as reflected by the overall percentage of optimal versus suboptimal codons of any given mRNA (*Presnyak et al., 2015*). Our data showing that the LARP4 CRD confers major instability despite its length of only 10% of the ORF, together with our analysis of its unusual codon bias including density of clusters of suboptimal codons (*Figure 3d*) argue that the CRD is a potent modular element of instability. These findings suggest that codon control of mRNA stability may reside in distinct regions of higher eukaryotic mRNAs. However, the extent to which this may be so and/or the types of subsets of mRNAs involved, if any, may be determined by future studies. The LARP4 CRD system, approaches and tools developed here should be useful toward addressing these issues.

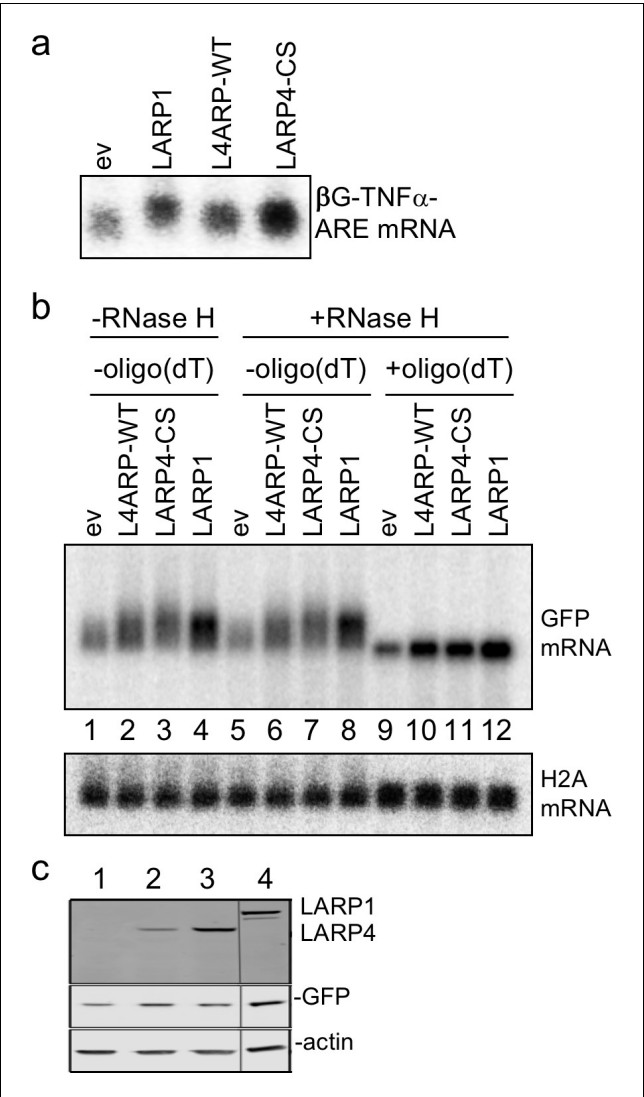

**Figure 9.** LARP1 promotes PAT 3' length stabilization of non-5'TOP mRNAs. (**a**) Northern blot after transfection of HEK293 cells with expression plasmids indicated above the lanes, and the βG-TNFα-ARE reporter. (**b**) Same as in a but transfected with GFP followed by RNase H assay in presence or absence of oligo(dT) probed for GFP mRNA as indicated. The blot in b was probed for histone H2A. (**c**) Western blot showing LARP proteins expressed.
DOI: https://doi.org/10.7554/eLife.28889.016

## Cellular tRNA levels provide insight into codon effects in higher eukaryotes

Another outstanding issue in codon optimality is the potential role of the cellular tRNA pool and codon-anticodon dynamics (*Chen and Shyu, 2017*). This has been challenging in higher eukaryotes because unlike in yeast, tRNA gene copy numbers do not correlate with codon use by efficiently translated mRNAs (*dos Reis et al., 2004*). This may reflect that significant numbers of tRNA genes of variable identities are inactive in different mammalian cell types (reviewed in *Orioli, 2017*). Our data demonstrated that the LARP4 CRD is a determinant of mRNA instability via its codon-specific match to HEK293 cell tRNA levels and their codon-anticodon dynamics.

We analyzed several full length LARP4 expression constructs that differ only in the synonymous codon composition of the CRD region. It was informative to consider effects of synonymous CRD substitutions on *LARP4* mRNA expression in conjunction with cellular tRNA levels and their codon-anticodon dynamics. This revealed that codons that must be decoded by very low abundance tRNAs

are concentrated in the LARP4 CRD along with a heavy bias of weak, all-A+U codons that must be wobble decoded, and clusters of these (*Figure 3d*). Two relevant findings are noteworthy. First, the CRD codon composition in conjunction with the HEK293 tRNAs predict that certain synonymous swaps cannot significantly improve optimality because all of the tRNAs for that amino acid are very low abundance. Specifically, all the tRNAs Thr, Pro and Ile for all Thr, Pro and Ile codons, are in bin-1 (*Table 1*). Therefore, the Thr, Pro and Ile codon synonymous swaps to a different tRNA anticodon would have only limited effect toward increasing *LARP4* mRNA levels, as was observed for construct CSc which contains 13 synonymous substitutions and collectively increased levels 2.4-fold.

The other noteworthy finding resulted from codon swaps of weak, all-A+U wobble codons to their stronger synonymous codons with a C in the third position. In these cases the new codon is decoded by the same tRNA but using a direct match anticodon wobble G34. These data revealed that wobble decoding of weak codons can be a significant determinant of suboptimality (*Figure 3c*). This latter point was demonstrated by comparing constructs CS-B and CS-Tyr that differ only in the third positions of all seven Tyr codons (UAU vs. UAC) in the CRD (*Figure 3c*), both of which are decoded by tRNA$^{Tyr}$GUA. Also, LARP4-CSb differs from LARP4-WT in 14 U-to-C synonymous codons that are wobble decoded in -WT but decoded by C:G codon:anticodon base pairs in -CSb, and increased expression 5-fold (*Figure 3c*).

Our analysis revealed that tRNA$^{Thr}$UGU led to increased protein production from LARP4-WT without increasing the mRNA levels, whereas some of the CS CRD constructs responded with tRNA dose-dependent increase in mRNA levels. This suggests that translation dependency and the instability component of the CRD can be uncoupled and that reversing the latter may require more than an increase in the levels of a single tRNA.

## tRNA$^{Thr}$UGU and other tRNAs are limiting in HEK293 cells

A significant component of this study was quantitative sequencing of HEK293 cell tRNAs and development of ctAIs. This led to the finding that the tRNAs for all Thr, Pro and Ile codons are of very low abundance, in bin-1 (*Table 1*). A striking finding was demonstration that over expression of the lowest abundance, tRNA$^{Thr}$UGU, increased production of LARP4 and GFP from transfected plasmids (*Figures 3g–h* and *5b*). As expected, this was accompanied by a shift of the corresponding mRNAs to heavier polysome fractions, presumably reflective of greater ribosome occupancy (*Figure 6g–i*). This was a robust activity that was specific since over expression of other low (or high) abundance tRNAs including tRNA$^{Thr}$CGU or tRNA$^{Tyr}$GUA did not increase LARP4 or GFP production (*Figure 3e,f* and data not shown). The responsiveness of GFP mRNA to tRNA$^{Thr}$UGU was remarkable (*Figure 6b–e,i*). We note that 14 of the 15 threonines in the codon-optimized GFP construct (*Haas et al., 1996*) are encoded by ACC codons and that these require wobble decoding because there is no tRNA with a GGU anticodon (*Table 1*) (*Gogakos et al., 2017*). While the wobble base modification status of human tRNA$^{Thr}$UGU has not been reported, in yeast it carries the ncm$^5$U modification (*Johansson et al., 2008*). This same ncm$^5$U modified base on tRNA$^{Pro}$UGG has been shown to wobble decode the Pro codon with C in the wobble position (*Johansson et al., 2008*). Understanding the determinants of responsiveness of a mRNA to tRNA$^{Thr}$UGU in this system will be a goal of future investigations.

## Does the CRD operate via a mRNA quality control mechanism?

While there is a large difference in the instability of the LARP4-WT and LARP4-CS-Tyr mRNAs as the latter accumulates to ~12 fold higher than the former (*Figure 3c*), we note that we do not know the source, *in vivo* kinetics, cell biology, specific factors involved nor the mechanism by which the CRD mediates the effect. Although our data on CRD codon composition and cognate tRNA levels would suggest that the mechanism is linked to slow translation (*Radhakrishnan and Green, 2016*), some points are nonetheless noteworthy. As was evident from our early analysis of *Figure 1e–g*, the different LARP4 truncation and deletion construct mRNAs containing or lacking the CRD would appear to have similar apparent translation efficiencies because the relative amounts of the mRNAs more or less match the relative amounts of proteins produced (*Figure 1e–g*). Moreover, this trend of match between relative amounts of mRNAs and proteins produced was maintained by the synonymous CS constructs analyzed in *Figure 3a–c* (and data not shown). It is also interesting to consider that although LARP4-WT mRNA accumulates to ~12 fold lower levels than LARP4-CS-Tyr mRNA, the

LARP4-WT mRNA that does survive appears to exhibit translational efficiency that is only modestly lower than LARP4-CS-Tyr mRNA based on polysome profile distributions (*Figure 6b–e*). This is not inconsistent with the observations supporting the generally similar apparent translation efficiencies of CS construct mRNAs made above. The data are consistent with some type of quality control mechanism that leads to decay of a large fraction of the LARP4-WT mRNA because it has suboptimal codons (*Radhakrishnan and Green, 2016*) (see *Brule and Grayhack, 2017*).

To gain insight into mechanism, we performed *in vitro* translation in extracts made from our HEK293 cells programmed with 7$^m$G capped and polyadenylated mRNAs synthesized by T7 RNA polymerase (*Rakotondrafara and Hentze, 2011*). Using equal amounts of synthetic transcripts corresponding to LARP4 WT and CS-Tyr mRNAs we sought to observe evidence of ribosome stalling in the CDR in the form of transiently arrested nascent polypeptides. After preliminary experiments revealed no difference in production of the polypeptides from the LARP4-WT and -CS-Tyr mRNAs in standard reactions, we performed time courses to more carefully focus on transition through the CRD region, codons 286–358. For these experiments we compared LARP4-WT(1-358) and LARP4-CS-Tyr(1–358) fragments because *in vivo* analysis showed that the WT CRD in the LARP4(1–358) construct was highly active (*Figure 1e*) and because this approach facilitated the *in vitro* analysis. However, there was comparably robust translation through the CRD regions of both of the synthetic mRNAs (*Figure 6—figure supplement 1*). Additional attempts to elucidate a difference in the *in vitro* translation of the two mRNAs by decreasing the temperature of the reactions were unsuccessful. Although the polysome distributions of LARP4-WT and LARP4-CS-Tyr (*Figure 6*) were consistent with lower translational efficiency of WT, we do not know the degree to which such a difference might be expected to manifest as a difference in these *in vitro* translation reactions. The *in vitro* translation results suggest among other things the possibility that the mechanism controlling the synonymous codon-specific differential expression/decay of LARP4-WT and -CS-Tyr in cells is coupled to transcription or another nuclear event(s) or process that is not faithfully executed during *in vitro* translation in extract of synthetic mRNAs. Such a possibility would be consistent with a quality control mechanism that is operational *in vivo*.

## Proposed mechanism of mRNA PAT 3' protection by LARP4

Prior to this work, a mechanism by which LARP4 or 4B may function in mRNA metabolism was unknown. We also note that sensitivity of single stranded poly(A) RNA to 3' end binding by LARP4 had remained untested. We used *in vitro* RNA binding to reveal that LARP4 exhibits sensitivity to the 3' end of poly(A) in a manner similar to that of the RNA binding pocket of La protein during RNA 3' end sequestration (*Teplova et al., 2006*; *Kotik-Kogan et al., 2008*). We note that this apparent similarity occurs despite other significant differences in RNA recognition by La and LARP4 including oligo(U) vs. oligo(A) specificity, RNA length requirements (*Yang et al., 2011*), and that the molecular and structural bases of these for LARP4 are unknown (reviewed in *Maraia et al., 2017*). Nonetheless, this 3' end sensitivity is consistent with a proposed mechanism for LARP4 for mRNA PAT 3' end protection from deadenylation. In any case, future experiments toward understanding how the PABP-interaction domains of LARP4 cooperate with its La module will be necessary to better understand its activity for PAT lengthening. In the current working model, the La module of LARP4 would bind poly(A), and its PBM and PAM2 would bind PABP, the latter of which also binds the PAT. Presumably, the LARP4 PAM2 would serve to compete with or displace from PABP, the PAM2-containing deadenylases which function as 3' exonucleases.

## LARP4 as a general factor for mRNA 3' PAT homeostasis

MEFs derived from LARP4 gene-deleted KO embryos created for this study were shown to bear RPmRNAs with shorter PAT length and faster decay than in WT MEFs. The mRNA-PAT length maintenance mediated by LARP4 characterized here was mostly for the highly abundant RPmRNAs. However, we wish to emphasize that LARP4 activity to increase PAT length was not limited to these mRNAs. Elevation of cellular LARP4 levels in response to increase in limiting tRNA or other means led to PAT lengthening and stabilization of βG-TNFα-ARE mRNA whose ARE directs deadenylation. Yet LARP4 would appear to differ from other mRNA stabilizing proteins which generally target lower abundance and transiently expressed mRNAs that are relatively unstable in their basal state (as compared to RPmRNAs) and regulated through their 3'UTRs, e.g., via AREs that indirectly modulate PAT

metabolism via trans-acting factors (*Chen and Shyu, 2017*). According to the model derived from our data, LARP4 differs because it binds directly to poly(A), the RPmRNAs are abundant and contain relatively very short 3' UTRs that are generally believed to be non-regulatory in the conventional sense. Thus, LARP4 would appear to be a general factor that is more directly involved in PAT length maintenance. The RPmRNAs comprise a substantial fraction of cellular mRNA in proliferating cells, and are critical and tightly regulated, including under growth control.

## LARP1 and LARP4B also promote mRNA PAT 3' length stabilization

Our data demonstrated mRNA PAT net lengthening activity for two other La module proteins, LARP4B and LARP1 (*Figure 4 and 9*), the latter of which is a regulator of ribosome biogenesis and a pro-cancer protein (*Aoki et al., 2013*; *Tcherkezian et al., 2014*; *Fonseca et al., 2015*; *Lahr et al., 2017*). By contrast, neither over expression of La protein, a significant fraction of which is cytoplasmic, nor LARPs 6 or 7, exhibited this activity even when accumulated at higher levels than LARP4-WT and -CS-R (data not shown). LARP4B contains a PAM2 and a separate PBM (*Bayfield et al., 2010*; *Schäffler et al., 2010*), and recent data identified a PAM2 candidate in LARP1 (*Fonseca et al., 2015*). LARP1 is a central factor in the regulation of translation of RPmRNAs in response to nutrition-related signals by controlling their repression which is mediated by its DM15 domain which binds their 5'TOP motif (*Lahr et al., 2017*). However, a recognized part of LARP1 activity in regulation is RPmRNA stabilization (*Tcherkezian et al., 2014*; *Fonseca et al., 2015*) and some data suggested that this may be mediated via binding to the 3' terminus of the PAT (*Aoki et al., 2013*). Our data showed that LARP1 stabilized/increased accumulation of βG-TNFα-ARE mRNA with accompanying mobility shift and it similarly shifted GFP mRNA which was shown to result from PAT lengthening (*Figure 9*). As neither of these mRNAs bear a 5'TOP motif, the data suggest that the PAT-mediated mRNA stability may be a separate activity of LARP1. This further suggests that PAT-mediated mRNA stability by LARP1 may be uncoupled from translation repression including when LARP1 is over-expressed as occurs in certain cancers and associates with non-TOP mRNAs (*Mura et al., 2015*; *Stavraka and Blagden, 2015*; *Hopkins et al., 2016*). However, while LARP1 contributes to RPmRNA translation as a central negative regulator, the present work suggests that LARP4 may be more of a constitutive positive factor in the control of RPmRNA homeostasis.

Finally, an intriguing aspect of this work is that tRNA availability might control or tune LARP4 activity levels. Because this could impact RPmRNA translation, it could theoretically represent signaling of tRNA availability to ribosome biogenesis.

# Materials and methods

## Cell lines and culture

HeLa Tet-Off (Clontech) and HEK293 were maintained in DMEM plus Glutamax (Gibco) supplemented with 10% heat-inactivated FBS (Atlanta Biologicals) in a humidified 37°C, 5% $CO_2$ incubator. Cultures were passed every 2–3 days. The HeLa Tet-Off cells were a gift obtained directly from Gerald Wilson (U Maryland, Baltimore). HEK293 cells were obtained from Tazuko Hirai in Bruce Howard's laboratory at the NIH. Our HEK293 and HeLa cell DNAs were both authenticated by the ATCC via STR profiling. HeLa Tet-Off cells are not commonly misidentified cell lines as listed by the International Cell Line Authentication Committee. Standardized mycoplasma testing (ATCC) was performed and tested negative.

## Mice

All mouse studies were performed at the NIH under protocol ASP 10–005 and approved by the IACUC of NICHD. The targeting vector HTGR06019_A_6_DO1 was generated by the trans-NIH Knock-Out Mouse Project (KOMP) and obtained from the KOMP Repository (www.komp.org). After exon 5 is floxed out, a termination codon will be encountered 16 codons following the La motif. The vector was linearized with AsiSI and electroporated into mouse embryonic stem cells. Neomycin-resistant colonies were isolated and scored for homologous integration by PCR amplification. Targeted clones were injected into C57BL/6 blastocysts and chimeric founder mice crossed with C57BL/6 females to establish the *Larpfloxed^{floxed+Neo}* line. Mice heterozygous for this *Larp4^{floxed}*

allele were crossed with *EIIa-cre* transgenic females to remove *Larp4* sequences between the 2 *loxP* insertions. No backcrossing was performed. All mice are maintained in microisolator caging within ventilated racks (Lab Products). Caging systems are changed once a week; cage tops and wire lids are changed every other week. The mice are fed NIH Autoclavable Rodent Chow. Mice are given chlorinated water in water bottles.

## DNA constructs

cDNA was generated from HeLa cells and used to amplify the coding regions of human La, LARP4, LARP4B, LARP6 and LARP7 starting at the second codon. The PCR products were cloned into the HindIII and BamHI sites of the pFLAG-CMV2 vector (Sigma-Aldrich). LARP4 truncation constructs were derived from the full length Flag-LARP4 vector. A deletion construct lacking codons 287–358 was purchased from Genewiz and subcloned into HindIII and BamHI of pFLAG-CMV2. Each construct was confirmed by sequencing.

Full length LARP4 with codons 287–358 of LARP4-WT swapped for synonymous codons (CS-I, CS-B, CS-W and CS-R, CSb, CSc) were obtained from Eurofins and subcloned into the HindIII and BamHI sites of the pFLAG-CMV vector. The LARP4 4.2 kb 3'-UTR was amplified by PCR from HeLa DNA using primers KpnI-UTR-Fwd 5'TAGGGCGGTACCAAAACAACAAAACTATTC-AAAAACTTCAC and XmaI-UTR-Rev 5'AATGTACCCGGGTTTTTTTTTTTTTTTTTTTTTTCTGCTTTTTAATAATTTTA TTTTTTTTCTAATTTTGTTAATTTCCCATAGCACC. The LARP4 WT or CS coding region was amplified using HindIII in the forward primer and KpnI in the reverse primer. After restriction digestion with KpnI, HindIII and XmaI, the CDS and 3'UTR were ligated and inserted into HindIII and XmaI sites of pFlag-CMV2 vector (Sigma-Aldrich).

Constructs containing 3 copies per plasmid of each tRNA gene for TyrGUA, PheGAA, ThrUGU or ProAGG, in pUC57-Kan were obtained from Genewiz. For each tRNA copy, 150 nt upstream and 90 nt downstream genome sequence was included. Codons 329(V) through 393(K) of LARP4B were replaced by codons 287–358 of LARP4 (WT or the CS-B sequence, were obtained from Genewiz and subcloned into HindIII and BamHI sites of pFlag-CMV2 vector (Sigma-Aldrich).

The pTRERβ-wt, encoding a rabbit β-globin minigene under control of a tetracycline-responsive promoter (also known as βG-wt) and pTRERβ-TNFα-ARE (containing the 38 nt ARE from TNFα, inserted into the β-globin 3'UTR) were a gift from G. Wilson (*Fialcowitz et al., 2005*). The pTRERβ-wt contains a unique BglII site located downstream of the stop codon of β-globin. Into this BglII site we cloned the LARP4 CRD sequence (corresponding to nucleotides for codons 287 to 358 plus TGA), to generate construct βG-stop-CRD (CRD in 3'-UTR). To generate βG-CRD-stop (CRD in CDS), we inserted the LARP4 CRD sequence in frame just before the stop codon. Another construct, containing a +2 frameshift in the CRD sequence just before the β-glo stop codon was obtained from Genewiz. In this construct, two As were inserted before the CRD sequence produce the +2 frameshift and nucleotide mutations were introduced downstream to convert premature stop codons to sense codons. β-globin-TNFα−ARE sequence from pTRERβ-TNFα-ARE was subcloned into NheI and PmeI sites of pcDNA3.1(-) to be expressed in HEK293 cells from a regular CMV promoter.

## Antibodies

Used in this study were anti-FLAG (Sigma, F1804), anti-GFP (Santa Cruz, sc-8334), anti-actin (Thermo Scientific, PA1-16890) and anti-GAPDH-HRP (Sigma Aldrich). Rabbit anti-LARP4 and anti- polyclonal rabbit antibodies were described (*Yang et al., 2011*; *Mattijssen and Maraia, 2015*).

## DNA transfection

All plasmids were verified as intact supercoiled and used in parallel at the same concentrations as determined by nanodrop OD260/280 and compared by ethidium bromide staining after gel electrophoresis (not shown). $5.5 \times 10^5$ HEK293 cells were seeded per well in 6 well plates one day prior to transfection with Lipofectamine 2000 (Invitrogen). Typically, 7.5 ul transfection reagent was used per well to transfect 2.5 ug of the pCMV constructs, plus 100 ng pcDNATPGFP plasmid (*Hogg and Goff, 2010*) and 100 ng pVA1 (*Maraia et al., 1994*) according to manufacturers' instructions. 24 hr after transfection, cells were split over multiple plates. To isolate protein samples, cells were washed with ice-cold PBS and cell lysis was directly into RIPA buffer (Thermo Scientific) containing protease

inhibitors (Roche). For RNA isolation, either the Maxwell 16 simply RNA cells kit (Promega) or Tripure (Roche) was used.

## Northern blotting

For mRNA and VA1 detection, total RNA was separated in 1.8% formaldehyde agarose gel and transferred to a GeneScreen-Plus membrane. For tRNA detection, total RNA was separated on 10% TBE/urea/polyacrylamide gels (Thermofisher) before transfer to a GeneScreen-Plus membrane (PerkinElmer) using iBlot Dry Blotting System (Invitrogen). The membranes were UV-cross-linked and vacuum-baked at 80°C for 2 hr. The sequences of oligo probes and their hybridization incubation temperatures (Ti) can be found in *Supplementary file 1*. Membranes were prehybridized in hybridization solution (6 x SSC, 2 x Denhardt's, 0.5% SDS and 100 ug/ml yeast RNA) for one hour at Ti. Hybridization of oligo probes was overnight at Ti.

## RNA binding

by recombinant purified LARP4(1–286) was as described (*Yang et al., 2011*). Analysis and quantitation was done using ImageQuant TL (GE Healthcare).

## β globin reporter mRNA half-life determinations

For experiments in 6-well plates, Lipofectamine 2000 (Invitrogen) was used to transfect HeLa Tet-off cells with 100 ng pTRERβ (or derivatives, b-globin under a Tet-responsive minimal CMV promoter, see 'DNA constructs'), 100 ng of a GFP-expression vector pcDNATPGFP containing a conventional CMV promoter (*Hogg and Goff, 2010*) according to the manufacturer's instructions. In some experiments, pCMV2 vectors, containing LARPs were co-transfected. Since LARP6 accumulates to relatively high levels, only half the amount of LARP6 plasmid was transfected compared to LARP4 WT and CS-R (amount adjusted with empty vector). The next day, cells were equally divided into multiple wells. 48 hr post transfection, media was replaced by media containing 2 µg/ml doxycycline (Sigma). For total RNA extraction with the Maxwell 16 simply RNA cells kit (Promega), cells were washed with PBS and directly lysed in homogenization buffer containing thioglycerol (Promega).

## Polysome profile analysis

was done by standard methods as described (*Mattijssen and Maraia, 2015*) using a programmable density gradient fractionation system spectrophotometer (model Foxy Jr.; Teledyne Isco, Lincoln, NE). $45 \times 10^5$ HEK293 cells were seeded in 10 cm culture plates so that they were 80–85% confluent after 16 hr. The cells were transfected with 816 ng GFP plasmid plus 10.2 ug pCMV2 plasmid containing either LARP4-WT or -CS-Tyr and 24.5 ug empty pUC19 or pUC19 containing 3 copies of the $tRNA^{Thr}UGU$ gene. The day after transfection, the cells from each plate were divided into two 15 cm culture plates (an aliquot was seeded into a 6-well plate for protein isolation the next day). One day later, fresh sucrose solutions (47% and 7%, wt/vol) in 10 mM HEPES, pH 7.3, 150 mM KCl, 20 mM $MgCl_2$, 1 mM DTT were prepared, filter sterilized and used to make the gradients with a Gradient Master (Biocomp). The cell growth medium was replaced 3 hr before addition of cycloheximide (Chx) at a final concentration of 100 µg/ml (from fresh made 10 mg/ml stock in water). After 5 min at 37°C, the cells were moved to ice and washed twice with ice-cold PBS plus 100 µg/ml Chx. Five ml of ice-cold PBS with 100 µg/ml Chx was added per plate, the cells were scraped and added to an ice-cold tube. The cell suspension was centrifuged for 3 min at 1,200 rpm at 4°C and the pellet taken up in 300 µl lysis buffer (10 mM HEPES, pH 7.3, 150 mM KCl, 20 mM $MgCl_2$, 1 mM DTT, 2% NP-40, 2x EDTA-free protease inhibitors (Roche), 100 µg/ml Chx, and 40 U/ml RNaseOUT (Invitrogen)) and kept on ice for 2 min with occasional flicking. The lysate was cleared by centrifugation at 13,000 rpm for 5 min at 4°C. Four hundred microliters of the gradient was removed from the gradient tubes, and the equivalent amount of 10 $OD_{260}$ units of each lysate was carefully loaded on top. The gradients were spun in an ultracentrifuge (Beckman SW41 rotor) at 33,000 rpm for 2 hr and 50 min at 4°C. One ml fractions were collected and RNA was purified from 500 µl of each fraction using the Maxwell 16 LEV simplyRNA kit with the Maxwell 16 instrument, which includes treatment with DNase I (Promega).

## tRNA-HySeq

tRNA-HySeq was as described (*Arimbasseri et al., 2015*; *Arimbasseri et al., 2016*). Briefly, cells were cultured under standard conditions (DMEM with 10% serum, 1X pencillin-strepomycin at 37°C with 5% CO2). Total RNA was isolated from near confluent cells using TriPure reagent (Roche) and resolved on a 6% polyacrylamide-urea-TBE gel. tRNA size (shorter than 5S rRNA were gel purified by incubating the crushed gel pieces in 0.3 M NaCl overnight. The tRNA was precipitated, quantified and 300 ng subjected to partial hydrolysis in 10 mM bicarbonate buffer pH 9.8. The hydrolyzed RNA was dephosphorylated using Calf-Intestine Alkaline Phosphatase (NEB) and the 5' termini were phosphorylated using $\gamma-^{32}$P-ATP and T4 polynucleotide kinase (NEB). Barcoded pre-adenylated 3' adapters were ligated to the 3' ends of the fragments (for sequences of all adapters and primers used see *Hafner et al., 2012*). In a parallel control reaction, two $^{32}$P-RNA size markers of 19 and 35 nt were also ligated to the adapters. The ligated fragments were resolved in a 10% polyacrylamide-urea-TBE gel alongside the adapter ligated size markers. tRNA fragments that migrated between the ligated 19 and 35 nt markers were isolated from the gel and subjected to 5' adapter ligation. The RNA fragments that with both 3' and 5' adapters were again size selected and subjected to reverse transcription (Superscript III, Invitrogen) at 42°C with the reverse primer. The RT products were subjected to limited PCR amplification and sequenced on an Illumina HiSeq 2500. The reads were analyzed as described earlier (*Arimbasseri et al., 2015*; *Arimbasseri et al., 2016*).

## Cellular tRNA index (ctAI)

Method for calculating ctAI is largely based upon that for calculating tAI (*dos Reis et al., 2004*) substituting tRNA read counts for tRNA gene copy numbers when calculating absolute adaptiveness values (W).

$$W_i = \sum_{j=1}^{n_i}(1-s_{ij})tRC_{ij}$$

Where $n_i$ is the number of tRNA isoacceptors which recognize codon i, $tRC_{ij}$ is the mapped mature read count for tRNA j which decodes codon i, and $s_{ij}$ is a movable constraint on the efficiency by which the decoding of i by j can occur. Values for $s_{ij}$ were optimized for expression data of constructs and are as follows for codon3:Anticodon1; U:G = 0.6, C:I = 0.3, G:U = 0.8, A:I = 0.5 and A:G = 0.9999. The $s_{ij}$ values are used here in the same way as described (*dos Reis et al., 2004*) $(1-s_{ij})$. The ctAI approach differs from tAI in notable respects. It attempts to optimize mRNAs for match to the existing tRNAs within the isoacceptor pool. As such, the relative adaptiveness value $w_i$ of a codon is weighted not against all codons, but within isoacceptor groups.

$$w_i = W_i/W_{iMax}$$

Where *i* represents a codon being assayed, and *iMax* represents the maximum isoacceptor *W* value for the group or 80 k reads (which is roughly 75%, i.e., the 3rd quartile), whichever is larger. Thus, ctAI is not an absolute overall measurement of translational strength across all possible constructs, but rather a measurement of codon optimization based on tRNA availability. As before for tAI, the ctAI is calculated for a mRNA (g) by:

$$ctAI_g = \left(\prod_{k=1}^{l_g} w_{i_{kg}}\right)^{1/l_g}$$

Where $i_{kg}$ is the k$^{th}$ codon of the mRNA in gene g of codon length $l_g$. As a result, ctAI measures the adaptation of codon selection in a mRNA to the observed tRNA pool.

## RNase H assay

Total RNA was isolated from HEK293 cells 48 hr after transfection. Two ug total RNA was diluted in a total volume of 11.5 ul H2O, then 4.5 ul of 4X hybridization buffer (40 mM Tris pH 7.5, 200 mM NaCl) and either 2 ul H2O or 2 ul oligo-dT20 (50 uM, Invitrogen) was added. Samples were heated at 85°C for 5 min then put in a 42°C water bath which was allowed to cool to 32°C (~1 °C/minute). 2 ul of 10X RNase H reaction buffer was added and 10 ul of RNase H (0.001 U/ul, Thermo Scientific)

and incubated at 37°C for one hour. Reactions were stopped by addition of 1.5 ul 0.5 M EDTA. To precipitate RNA, 2 ul glycoblue and 13.4 ul 3 M NaAc pH 5.2 were added and mixed followed by 375 ul EtOH and incubation at −80°C for one hour. Samples were spun for 30 min at 13,000 rpm at 4°C and RNA pellets washed with 1 ml 75% EtOH. RNA was analyzed on northern blot after formaldehyde agarose gel electrophoresis.

## Mouse embryonic fibroblasts (MEFs)

MEFs were generated from E14.5 embryos from the same litter by standard methods. Each MEF cell line was derived from a different embryo. MEFs were derived from KO and WT matched siblings, all females. MEFs at passage 3 were transfected with SV40 Large-T antigen-expressing plasmid pBSSVD2005 (Addgene plasmid 21826) using Lipofectamine 2000 (Invitrogen) and subcultured at 1:10 for at least 5 passages. Cells were maintained in DMEM plus Glutamax (Gibco) supplemented with 10% heat-inactivated FBS (Atlanta Biologicals) in a humidified 37°C, 5% $CO_2$ incubator. Cultures were passed every 2–3 days.

## *In vitro* translation

HEK293 cell lysate was prepared from cells at 70% confluency by adding an equal volume of lysis buffer (10 mM HEPES pH 7.3, 10 mM KAc, 0.5 mM MgAc, 5 mM DTT and protease inhibitors (Roche) to a PBS-washed cell pellet and incubating for 45 mins on ice. The cells were then passed 10 times through a 30.5 G needle and checked under a microscope for a lysis of >60%. The lysate was spun at 14,000 g for 1 min to remove debris and nuclei. The supernatant was aliquoted and immediately frozen at −80C. DNA templates for T7 RNA polymerase-mediated transcription, were generated by PCR to obtain the following fragments: Flag-LARP4-1-286 (to mark the start of the CRD) and 2 versions of Flag-LARP4- 358 (end of the CRD), a WT version and CS-Tyr. Using the *mMES-SAGE mMACHINE* T7 Ultra Kit (Thermofisher), 7$^m$G 5' capped and 3' polyadenylated mRNAs were generated. Polyadenylation after addition of PolyA polymerase was confirmed by denaturing gel electrophoresis (not shown). The *in vitro* translation reaction contained the following in 10 ul: 40% cell extract, 50 mM KAc, 2.5 mM MgAc, 20 U Superasin, 200 ng mRNA template, 1.6 mM HEPES pH 7.3, 2 mM creatine phosphate, 0.01 ug/ul creatine kinase, 10 uM spermidine, 10 uM amino acid mix (no methionine), 10.2 uCi $^{35}$S-Methionine (Perkin Elmer)). Reactions were placed at 37C and after indicated times placed on ice and quenched by addition of 10 ul EDTA (25 mM final). The reactions were then subjected to immunoprecipitation using *Anti-FLAG* M2 Magnetic Beads (Sigma) according to manufacturer's protocol. Immunoprecipitated material was eluted from the beads using SDS buffer containing b-mercaptoethanol and heated for 5 mins at 80C. Samples were loaded on an SDS-PAGE gel, then blotted to nitrocellulose and imaged on a phosphorimager screen.

## Acknowledgements

We wish to thank Drs. K. Pfeiffer and A. Grinberg (NICHD) for their help with generation of ES cells and gene-altered mice, and B. Fonseca for the LARP1 plasmid, and A. Hinnebusch for discussion and comments on the manuscript. The authors declare that they have no competing financial interests. This work was supported by the Intramural Research Program of the *Eunice Kennedy Shriver* National Institute of Child Health and Human Development, NIH.

## Additional information

### Funding

| Funder | Grant reference number | Author |
| --- | --- | --- |
| Eunice Kennedy Shriver National Institute of Child Health and Human Development | HD000412-30 | Richard J Maraia |

The funders had no role in study design, data collection and interpretation, or the decision to submit the work for publication.

## Author contributions

Sandy Mattijssen, Conceptualization, Formal analysis, Validation, Investigation, Methodology, Writing—original draft, Writing—review and editing, Designed and performed most experiments including isolation and analysis of MEFs, PAT length analysis, CS construction, mRNA stability analyses, protein and mRNA expression analysis, isolated and analyzed MEFs, analyzed data and contributed to writing the manuscript; Aneeshkumar G Arimbasseri, Investigation, Methodology, Writing—original draft, Performed tRNA-HySeq experiments; James R Iben, Formal analysis, Methodology, Writing—original draft, Writing—review and editing, Analyzed tRNA-HySeq data, derived ctAI scores, developed sliding window score plot, and designed some CS constructs; Sergei Gaidamakov, Resources, Formal analysis, Methodology, Carried out protein purification, performed and analyzed in vitro RNA binding, and oversaw mouse breeding; Joowon Lee, Formal analysis, Investigation, Writing—original draft, Performed LARP subcloning, protein and mRNA expression analysis; Markus Hafner, Formal analysis, Investigation, Methodology, Writing—review and editing, Oversaw/supervised tRNA-HySeq data acquisition and analyzed data; Richard J Maraia, Conceptualization, Resources, Data curation, Formal analysis, Supervision, Investigation, Writing—review and editing, Oversaw/supervised data acquisition, designed experiments and CS constructs including the wobble constructs, analyzed CRD codon composition and sliding window plots, and contributed to writing the manuscript

## Author ORCIDs

Markus Hafner, https://orcid.org/0000-0002-4336-6518
Richard J Maraia, http://orcid.org/0000-0002-5209-0066

## Ethics

Animal experimentation: This study was performed in strict accordance with the recommendations in the Guide for the Care and Use of Laboratory Animals of the National Institutes of Health, under NICHD ASP# 10-005. All of the animals were handled according to approved institutional animal care and use committee (IACUC) of the NICHD.

## Decision letter and Author response

Decision letter https://doi.org/10.7554/eLife.28889.019
Author response https://doi.org/10.7554/eLife.28889.020

# Additional files

### Supplementary files

• Supplementary file 1. Table list of s oligonucleotide probe sequences and their hybridization incubation temperatures (Ti).
DOI: https://doi.org/10.7554/eLife.28889.017

• Transparent reporting form
DOI: https://doi.org/10.7554/eLife.28889.018

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
