## [Decision Letter]

Thank you for submitting your article "LARP4 mRNA codon-tRNA match contributes to LARP4 activity for mRNA poly(A) tail length protection" for consideration by *eLife*. Your article has been reviewed by three peer reviewers, one of whom, Nahum Sonenberg (Reviewer #1), is a member of our Board of Reviewing Editors, and the evaluation has been overseen by James Manley as the Senior Editor. The following individuals involved in review of your submission have agreed to reveal their identity: Maria Sasi Conte (Reviewer #2); Jack D Keene (Reviewer #3).

The reviewers have discussed the reviews with one another and the Reviewing Editor has drafted this decision to help you prepare a revised submission.

Summary:

Maraia and coworkers report the LARP4 RNA binding protein binds to poly A RNA and interacts with poly A binding protein through protein-protein interactions. The functions of LARP4 include the stabilization of the mRNAs bound to the poly A. However, the stabilization of targeted mRNAs of LARP4 depends on interactions between its mRNA coding region determinant and specific transfer RNAs. The variation of codon sequences of tRNA interaction in the CRD are tested for their ability to vary the stability and subsequent translation of the LARP4 mRNA, that in turn, regulates poly A length of targeted ribosomal protein mRNAs. This is a grand scheme of regulation that is entirely novel, and potentially powerful. The data are strongly supportive of the regulatory model demonstrating that LARP4 is at the center of posttranscriptional coordination of production of the ribosomal proteins in mammalian cells. The authors used many in vitro and in vivo (MEFs knockout of LARP4) approaches and many quantifiable molecular measures to support their conclusions.

Essential revisions:

1) The authors have presented a very thorough and detailed series of experiments that strongly support their interpretations of this complex biological system. However, It is challenging for the reader to work through the text, the figure legends, and abundant data. The authors should make an effort to simplify the text as much as possible.

2) The authors should provide further proof for tRNAThrUGU-dependent temporary ribosome stalling/translation slow down at the CDR segment of LARP4 mRNA. For example, they could perform ribosome profiling. Alternatively, or in addition, ribosome stalling at CDR could be examined by the generation of transiently arrested nascent polypeptides in a cell-free system of translation of Larp4 mRNA.

---

## [Author Response]

Essential revisions:1) The authors have presented a very thorough and detailed series of experiments that strongly support their interpretations of this complex biological system. However, It is challenging for the reader to work through the text, the figure legends, and abundant data. The authors should make an effort to simplify the text as much as possible.

We agree that a most critical improvement was to be text revision. This stems from the fact that the paper is complex, bridging two intricate stories, one of which is intrinsically complicated and not yet mainstream, effects of tRNAs and codons, and the other on mRNA poly(A) metabolism. While the terse style was not inappropriate for expert reviewers, it could be much improved toward helping readers better appreciate the paper and thank the reviewers for the opportunity to revise and thereby increase its accessibility. We have made text revisions throughout the manuscript toward this objective. Two types of modifications were made throughout; many text passages/descriptions were decondensed and simplified, and second, we added several new section headings to introduce and highlight, and provide more descriptive and logical flow between transitions.

2) The authors should provide further proof for tRNAThrUGU-dependent temporary ribosome stalling/translation slow down at the CDR segment of LARP4 mRNA. For example, they could perform ribosome profiling. Alternatively, or in addition, ribosome stalling at CDR could be examined by the generation of transiently arrested nascent polypeptides in a cell-free system of translation of Larp4 mRNA.

Ribosome profiling as an approach to provide further proof of tRNAThrUGU-dependent translation slow down as suggested is presently beyond our ability in a reasonable time frame for two reasons. One is overcoming the background of endogenous LARP4 mRNA sequence reads in the most important region, the CRD. We know from our work that in the experimental system used throughout, LARP4-WT levels from transfected plasmid accumulate at levels similar to endogenous LARP4 (see Yang et al., 2011). While transfected LARP4-WT mRNA can be discerned from endogenous by size/mobility on northern blots, we know from our RNA-seq data (a component of ribosome profiling) that their internal sequence reads cannot be distinguished, yet the endogenous would comprise about 50% of the signal. Also, endogenous LARP4 mRNA is subjected to a level of regulatory instability via its ARE-containing 3' UTR that may or may not confound analysis of the transfected construct. Thus an approach would be to knock down endogenous LARP4 mRNA but this would require separate shRNA plasmid or siRNA transfection preceding transfection of the multiple other plasmids, (LARP4 constructs to be examined, the tRNA plasmid, and control plasmid, GFP). We know that these HEK293 cells become sensitive to complex transfection regimens, and we have not worked out the siRNA or other conditions for such an experiment. Second is our lack of perfected technology; our actual ribosome profiling data have revealed that we have not yet solved the three nucleotide codon phase issue and do not yet provide high resolution mapping data.

Because we could not expect to resolve these two ribosome profiling issues in a reasonable time frame, we attempted to examine ribosome stalling in the CRD by the alternative approach suggested in essential revisions #2, i.e., hoping to generate paused/arrested nascent polypeptides in a cellfree translation extract programmed with LARP4 mRNA. We had not performed such experiments before, using translation competent extract from cells of interest, the HEK293 cells. We followed the method described by the Hentze lab (Rakotondrafara and Hentze, 2011), in vitro translation in our extract, programmed with 7mG 5' capped and 3' polyadenylated mRNAs synthesized by T7 RNA polymerase. After preliminary experiments revealed no difference between production of the corresponding LARP4-WT and LARP4-CS-Tyr polypeptides in the standard reactions, we examined time courses to focus on the time during which the CRD, codons 286-358, is translated. However, even then, we detected comparably robust translation through the CRDs of both of the synthetic mRNAs. We describe the results in more detail in the Discussion section of the revised manuscript and in Figure 6—figure supplement 1. We also note that lowering temperature did not reveal differences between LARP4-WT and LARP4-CS-Tyr. We appreciate that we could further try to find conditions such as titrating extract concentration and/or amino acids to show differential slow down in the WT CRD but this would also shift focus of the manuscript and we have not done so. We note in the Discussion: The in vitro translation results suggest among other things the possibility that the mechanism controlling the synonymous codon-specific differential expression of LARP4-WT and – CS-Tyr in cells is coupled to transcription or another nuclear event(s) or process that is not faithfully executed during in vitro translation in extract of synthetic mRNAs. Such a possibility would be consistent with a quality control mechanism. We believe that it is worthwhile to consider the results in the context of the fact that when forced to accumulate comparable amounts of mRNA by adjusting the amounts of transfected plasmid, the LARP4-WT and -CS-Tyr mRNAs produce comparable amounts of protein (Figure 6 lanes 1 and 3). This is more broadly considered in a new section in the Discussion part of the revised manuscript: **"**Does the CRD operate via a mRNA quality control mechanism?"